# UniVid: The Open-Source Unified Video Model

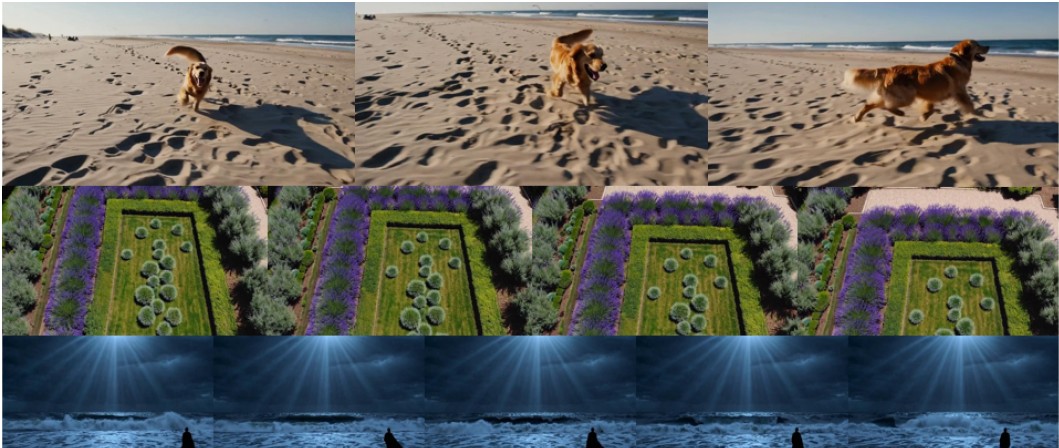

Figure 1: We present **UniVid**, an open-source unified video model for both understanding and generation tasks. Our model requires only a small amount of high-quality data for fine-tuning, achieveing competitive results across various tasks.

## ABSTRACT

Unified video modeling combining generation and understanding capabilities is increasingly important, yet faces two key challenges: maintaining semantic faithfulness during flow-based generation due to text-visual token imbalance and the suboptimality of uniform cross-modal attention across the flow trajectory, and efficiently extending image-centric MLLMs to video without costly retraining. We present **UniVid**, a unified architecture that couples an MLLM with a diffusion decoder through a lightweight adapter, enabling both video understanding and generation. We introduce *Temperature Modality Alignment* to improve prompt adherence and *Pyramid Reflection* for efficient temporal reasoning via dynamic keyframe selection. Extensive experiments on standard benchmarks demonstrate the state-of-the-art performance of our unified video model, achieving a **2.2%** improvement on VBench-Long total score compared to the previous SOTA method EasyAnimateV5.1, and **1.0%** and **3.3%** accuracy gains on MSVD-QA and ActivityNet-QA, respectively, compared with the best prior 7B baselines.

## 1 INTRODUCTION

Video intelligence encompasses two core capabilities: generation and understanding. Generation enables content creation, simulation, and data augmentation through diffusion and flow models (Shi et al., 2020; Podell et al., 2024; Wang et al., 2025; Blattmann et al., 2023a). Understanding powers perception, retrieval, analytics, and human-computer interaction via multimodal LLMs (Wang et al., 2024a; Chen et al., 2024c; Lin et al., 2024; Bai et al., 2025). Real-world applications increasingly demand unified systems that combine both capabilities within a single framework. Recent efforts toward unified video modeling have converged on two paradigms. The first is an autoregressive (AR)–centric route: all modalities (text, images, video) are projected into a shared discrete token space and a single Transformer is trained with next-token prediction over multimodal sequences; representative examples include Emu3 (Wang et al., 2024b) and Chameleon (Lu et al., 2023). The second is a hybrid diffusion–AR route: a multimodal AR backbone governs understanding and

control signals, while a diffusion video decoder renders high-fidelity frames from high-level visual tokens; recent works such as Transfusion (Zhou et al., 2024a) and Show-O (Xie et al., 2025a) follow this pattern. In this work, we adopt the hybrid route to retain high-quality rendering while leveraging an MLLM for semantic control and interpretability.

However, even within this hybrid setting, unified video modeling faces two key challenges. First, maintaining semantically faithful conditioning in video diffusion across the flow trajectory is difficult. Text prompts convey high-level intent but under-specify pixel-aligned details; in MM-DiT-style Esser et al. (2024) models, the cross-modal signal can be diluted by the numerical imbalance between few text tokens and many visual tokens, and the role of guidance is inherently timestep-dependent—early steps benefit more from strong semantic constraints, whereas later steps benefit from visual detail refinement, yielding prompt–video drift that worsens with longer, higher-resolution clips. Second, extending image-centric MLLMs to video faces two key challenges: the computational cost of temporal modeling (dedicated encoders, long-context handling, large-scale training) that risks destabilizing existing capabilities, and the mismatch between video's vast temporal information and the typically small subset relevant to any question. Traditional approaches either process all frames uniformly, causing inefficiency and noise, or use fixed sampling that may miss critical evidence. Furthermore, different question types demand different strategies—static questions need distinctive keyframes while dynamic questions require understanding temporal transitions.

To address these challenges, our motivation is twofold. First, on the generation side, we leverage multimodal understanding to construct structure-aware tokens in the language space that encode both global semantics and localized cues; these tokens are used as faithful semantic conditioning for a diffusion video decoder, and we schedule cross-modal attention over flow steps so that early integration emphasizes textual intent while later steps emphasize visual refinement. Second, on the understanding side, we develop an adaptive evidence selection approach that extends image-centric MLLMs to video without substantial architectural changes. This requires a mechanism that can iteratively explore and refine the evidence set based on feedback, balance exploration of new frames with exploitation of current evidence, and learn from failure signals to improve future selections. This suggests a sequential decision-making framework, but rather than traditional parameter updates, we implement a form of verbal test-time reinforcement learning. We develop Pyramid Reflection, where policy improvement occurs through natural language refinement—the Reflector verbally adjusts search queries based on feedback, while SigLIP2 (Tschannen et al., 2025) enables query-driven keyframe selection that iteratively expands or prunes the evidence set.

Hence, we propose **UniVid**, a unified architecture that couples a multimodal LLM with a diffusion video decoder via a lightweight conditioning adapter: the LLM ingests text and salient visual evidence and outputs rich semantic understandable tokens that both support reasoning and condition the decoder for text/image-to-video generation. To stabilize guidance in MM-DiT (Esser et al., 2024), we introduce *Temperature Modality Alignment*, a timestep-aware, temperature-adjusted cross-modal attention schedule that emphasizes semantic intent early and visual refinement late, mitigating text suppression and improving prompt faithfulness. To enable efficient understanding with minimal change, we introduce *Pyramid Reflection*, which implements sequential decision-making through SigLIP2-based keyframe selection and an Actor–Evaluator–Reflector loop that verbally adjusts search strategies while progressively expanding or pruning context. Through extensive evaluation on standard benchmarks, we validate the superior capability of our unified approach, which consistently outperforms existing methods across multiple video-centric tasks, demonstrating the potential of unified modeling for comprehensive video intelligence.

Our contribution can be summarized below:

- We introduce **UniVid**, a unified paradigm that couples an MLLM with a diffusion video decoder via a lightweight conditioning adapter; the MLLM produces rich, understandable semantic tokens that both support reasoning and condition text/image-to-video generation.
- We propose *Temperature Modality Alignment*, a timestep-aware, temperature-adjusted cross-modal attention schedule in MM-DiT that strengthens early semantic guidance and later shifts emphasis to visual refinement; we further develop *Pyramid Reflection* with SigLIP2-based keyframe selection to enable efficient temporal reasoning with minimal architectural change and training.
- We conduct comprehensive experiments on MSVD-QA (Piergiovanni et al., 2022), MSRVTT-QA (Piergiovanni et al., 2022), TGIF-QA (Jang et al., 2017), and ActivityNet-QA (Yu et al.,

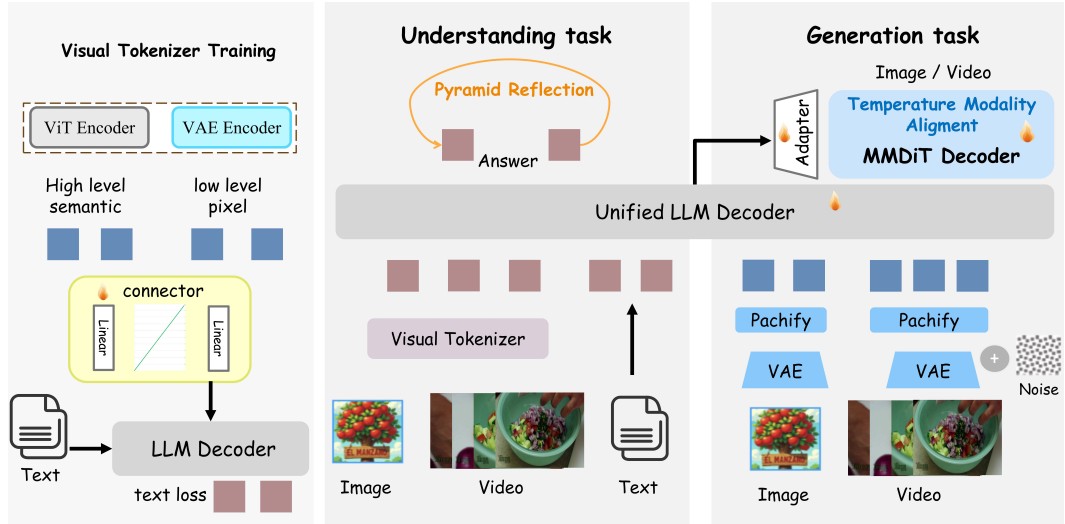

Figure 2: Overall architecture of our proposed UniVid for unified video understanding and generation. Notably, for the understanding task, we adopt only the ViT encoder to achieve a better efficiency–accuracy trade-off.

2018) for understanding, and on VBench for generation, demonstrating competitive performance and efficiency. Ablations verify the contribution of each component.

## 2 RELATED WORK

**Video generation.** Video generation has seen remarkable advancements with the rise of diffusion models and generative adversarial networks tailored for temporal data. Recent diffusion or flow based frameworks, such as Video Diffusion Models (Ho et al., 2022b), Imagen Video (Ho et al., 2022a), VideoCrafter2 (Chen et al., 2024a) and Stable Video Diffusion (Blattmann et al., 2023b), have produced high-fidelity clips with improved temporal consistency, enabling applications in creative generation and simulation (Liu et al., 2025; Shi et al., 2025). Latent diffusion techniques (Blattmann et al., 2023c) further improve efficiency by operating in compressed latent spaces, enabling scalable video generation. In parallel, GAN methods like MoCoGAN (Tulyakov et al., 2018) and StyleGAN-V (Skorokhodov et al., 2022) explore alternative formulations. Despite these advances, maintaining long-term temporal consistency in extended sequences remains challenging, as summarized by recent surveys and analyses (Melnik et al., 2024; Yin et al., 2025).

**Video understanding.** Recent progress in video understanding has been driven by transformer-based architectures and self-supervised learning paradigms that effectively model spatio-temporal relationships. Methods like MViT (Fan et al., 2021), Video Swin Transformer (Liu et al., 2022), TimeSformer (Bertasius et al., 2021) and ViViT (Arnab et al., 2021) have advanced the field by capturing long-range dependencies across video frames, achieving strong performance on datasets such as Kinetics-700 (Carreira et al., 2019). Beyond supervised training, self-supervised approaches—including masked modeling (VideoMAE (Tong et al., 2022), MaskFeat (Wei et al., 2022), OmniMAE (Girdhar et al., 2023)) and early contrastive methods (VideoMoCo (Pan et al., 2021))—leverage unlabeled videos to learn robust, transferable representations, reducing dependence on costly annotations and benefiting action recognition and video segmentation.

**Unified multimodal models.** Unified multimodal modeling has progressed from joint vision–language pretraining to architectures that support both understanding and generation across modalities. Foundational systems like CLIP (Radford et al., 2021) establish large-scale alignment, while BEiT-3 (Wang et al., 2023) and UnifiedMLLM (Li et al., 2024) broaden task coverage. Pushing toward unified generation, Show-o (Xie et al., 2025a) integrates autoregression with discrete diffusion within a single Transformer to support VQA, text-to-image, and various editing tasks. In a complementary direction focused on robustness rather than general any-to-any generation, FLUID (Cuong et al., 2025) uses token-level distillation for cross-modal fusion. Open generalist systems

then aim to unify understanding and generation end-to-end: BAGEL (Deng et al., 2025) offers an open, decoder-only framework with parallel language and diffusion branches trained jointly, achieving competitive results across image-centric tasks, and BLIP3-o (Chen et al., 2025) releases a fully open family where a diffusion transformer is coupled to strong multimodal understanding, yielding unified image understanding and generation. Extending unification from images to video, Omni-Video (Tan et al., 2025) teaches an MLLM to emit continuous visual tokens that are adapted and consumed by a diffusion video decoder, enabling generation, editing, and understanding in one pipeline.

# 3 THE PROPOSED METHOD

## 3.1 OVERVIEW

Our goal is a unified multimodal video model that supports both generation and understanding within a single framework. To this end, we adopt a three-stage hierarchical training recipe that first aligns the conditioning between the MLLM and the generator, then finetunes the MLLM and introduces Pyramid Reflection, which augments the understanding branch with temporal cues, and finally co-adapts both branches end-to-end. Fig. 2 presents the overall UniVid architecture.

## 3.2 ARCHITECTURE

**Multimodal archiecture.** The multimodal large language model serves as the core reasoning engine. Text inputs are processed through a standard tokenizer, while visual inputs follow different encoding paths depending on the target branch. For the generation branch, images are encoded using both ViT (Dosovitskiy et al., 2021) for semantic features and VAE (Kingma & Welling, 2019) for pixel-level details. For the understanding branch, only ViT encoding is employed, as video understanding tasks primarily rely on high-level semantic understanding rather than fine-grained pixel details. The encoded visual features are then projected into the textual token space and concatenated with text tokens, allowing the LLM to output unified multimodal representations.

**Generation branch.** The generation pathway employs a DiT-based model Wan 2.2 (Wang et al., 2025) conditioned on rich semantic representations extracted from MLLM outputs through a lightweight adapter. The system processes video generation in latent space using a 3D VAE (Zhao et al., 2024), with conditioning signals integrated via cross-attention mechanisms.

**Understanding branch.** For video understanding, multi-frame evidence is encoded by the ViT (Dosovitskiy et al., 2021) and fused with text; the LLM produces an initial textual answer. We then apply Pyramid Reflection, a query-driven, hierarchical loop that iteratively expands or prunes keyframe context via SigLIP2 (Tschannen et al., 2025) selection and refines the frame space via an Actor–Evaluator–Reflector process, yielding the final answer without modifying the backbone.

Conclusively, our generation builds on the MLLM's strong comprehension, while video understanding uses Pyramid Reflection to leverage the MLLM and collaborate with an LLM for efficient and accurate answers.

## 3.3 CONDITIONAL GENERATION WITH TEMPERATURE MODALITY ALIGNMENT

Given fused tokens from the understanding path, the MLLM output $Z_u$ is mapped to time-indexed conditions by a lightweight adapter $g_\phi$:

$$C_t = g_\phi(Z_u, t) \in \mathbb{R}^{M_t \times d_c}, \tag{1}$$

where $M_t$ is the number of conditioning tokens at timestep $t$ and $d_c$ is the conditioning dimension.

Let the 3D VAE define the latent trajectory $\{z_t\}$ along the flow, where $z_t \in \mathbb{R}^{H \times W \times F \times C}$ represents the latent representation with spatial dimensions $H \times W$, temporal frames $F$, and channels $C$. The Wan 2.2 DiT predicts the velocity field under cross-attention to $C_t$, then we integrate the probability–flow ODE to obtain $\hat{z}_0$, which the VAE decoder converts to video frames.

Inspired by TACA (Lv et al., 2025), we adapt its finding that text is suppressed in MM-DiT (Esser et al., 2024) because (i) the softmax over a much larger pool of visual tokens ($N_{\text{vis}} \gg N_{\text{txt}}$) dilutes

---

**Algorithm 1** Pyramid Reflection as Test-time RL

---

**Require:** video $V$, question $q$
1: Uniformly sample $N=64$ frames; *encode once and cache* visual embeddings
2: From 16 frames, summarize into a global caption $C_g$
3: Initialize state $s_1 \leftarrow (q, C_g, W=\varnothing)$, policy $\pi$ with mode router **expand/shrink**
4: **for** $r = 1$ to $R \leq 3$ **do**
5:     **Action**: $a_r \sim \pi(s_r)$
     **expand**: add frames most relevant to current search text
     **shrink**: prune to diverse key frames using cached similarities
6:     Update working set $W$ accordingly using cached embeddings          (index-only change)
7:     **Actor**: answer using ordered $W$ conditioned on $C_g$
8:     **Evaluator**: score $\hat{r}_r \in [0, 1]$ as confidence signal
9:     **if** $\hat{r}_r \geq \tau$ **then return** answer
10:    **elseReflector**: refine the search text $q \leftarrow$ short declarative cue
11:       Update state $s_{r+1} \leftarrow (q, C_g, W)$          (verbal policy improvement)
12:    **end if**
13: **end for**
14: **return** fallback answer from $C_g$

---

attention mass on text keys, and (ii) conditioning plays different roles across timesteps (early semantics, late detail). We therefore strengthen the visual-to-text path in Wan 2.2 (Wang et al., 2025) with a simple schedule:

$$\tilde{S}_{v \to t}(u) \;=\; \alpha_{\text{txt}}(u)\, S_{v \to t}, \qquad u \in [0, 1], \tag{2}$$

where $u$ is the normalized flow matching progress (0 early, 1 late), $S_{v \to t}$ denotes the visual-to-text attention scores, and $\tilde{S}_{v \to t}(u)$ represents the modulated attention scores. The modulation factor is defined as:

$$\alpha_{\text{txt}}(u) = \begin{cases} 1 + \dfrac{\lambda_{\text{txt}}}{2}\Big(1 + \cos\Big(\dfrac{\pi u}{0.4}\Big)\Big), & u \in [0,\ 0.4], \\ 1, & u \in (0.4,\ 1], \end{cases} \qquad \lambda_{\text{txt}} = 0.3. \tag{3}$$

Thus, text guidance is strongest early and decays to neutral ($\alpha_{\text{txt}} \to 1$) late, improving prompt faithfulness without over-constraining details.

For reference-image that requires identity stability, we apply a small late-stage boost to visual cross-attention:

$$\tilde{S}_{v \to v}(u) = \alpha_{\text{img}}(u)\, S_{v \to v}, \tag{4}$$

where $S_{v \to v}$ represents visual cross-attention scores and

$$\alpha_{\text{img}}(u) = \begin{cases} 1, & u \in [0,\ 0.6], \\ 1 + \dfrac{\lambda_{\text{img}}}{2}\Big(1 - \cos\Big(\dfrac{\pi(u - 0.6)}{0.4}\Big)\Big), & u \in (0.6,\ 1], \end{cases} \qquad \lambda_{\text{img}} = 0.3. \tag{5}$$

### 3.4 PYRAMID REFLECTION FOR UNDERSTANDING

**Formulation.** We cast video question answering as test-time reinforcement learning over a small, ordered evidence set. The state at round $r$ is $(s_r, W_r, C_g)$, where $s_r$ is a short search text, $W_r$ is an ordered subset of frames, and $C_g$ is a global caption distilled once from uniformly sampled seeds. The action is to reconfigure $W_r$ given $s_r$, either by adding frames (expand) or by pruning to a diverse core (shrink). The policy $\pi_s$ is a retrieval rule driven by text–image similarity and a diversity term; it maps $s$ to a distribution over frame indices. The environment returns an answer $a$ produced by the Actor and a scalar reward $r \in [0, 1]$ from the Evaluator. Policy improvement is carried out verbally: the Reflector emits a refined $s_{r+1}$ that concentrates on disambiguating cues such as before/after, first/last, motion phase, color, or role. The loop stops early when $r$ exceeds a confidence threshold.

**Policy class.** We instantiate $\pi_s$ with a cached-embedding retriever. All $N$ candidate frames are embedded once by a vision encoder; the text side uses $\phi(s)$. For expand we add the highest-scoring

Table 1: T2V performance on VBench-Long (Huang et al., 2024).

| Method | Overall Scores | | | Technical Quality | | | | | Aesthetic Quality | |
|---|---|---|---|---|---|---|---|---|---|---|
| | Total Score↑ | Quality↑ | Semantic↑ | Subject↑ | Background↑ | Temporal↑ | Motion↑ | Dynamic↑ | Aesthetic↑ | Imaging↑ |
| EasyAnimateV5.1 (Fu et al., 2024b) | 83.42 | 85.03 | 77.01 | 98.00 | 97.41 | 99.19 | 98.02 | 57.15 | **69.48** | 68.61 |
| MiniMax-Video-01 (MiniMax, 2024) | 83.41 | 84.85 | 77.65 | 97.51 | 97.05 | 99.10 | 99.22 | 64.91 | 63.03 | 67.17 |
| Kling 1.6 (Technology, 2025) | 83.40 | 85.20 | 76.99 | 97.40 | 96.84 | 99.64 | 99.13 | 62.22 | 64.81 | 69.70 |
| Wan2.1-T2V-1.3B (Wang et al., 2025) | 83.31 | 85.23 | 76.95 | 97.56 | 97.93 | 99.55 | 98.52 | 65.19 | 65.46 | 67.01 |
| Wan2.2-TI2V-5B (Wang et al., 2025) | 83.59 | 85.64 | 76.53 | 97.66 | 98.03 | 99.10 | 98.71 | 65.76 | 65.52 | 67.51 |
| HunyuanVideo (Kong et al., 2024) | 83.24 | 85.86 | 75.82 | 97.32 | **97.93** | 99.49 | 98.99 | **70.83** | 60.36 | 67.56 |
| Gen-3 (Runway, 2024) | 82.32 | 84.11 | 75.17 | 97.01 | 96.62 | 99.61 | 99.23 | 60.14 | 63.34 | 66.82 |
| Vchitect-2.0 (VEnhancer) (Fan et al., 2025) | 82.24 | 83.54 | 77.06 | 96.83 | 96.66 | 98.97 | 98.98 | 63.89 | 60.41 | 65.35 |
| CogVideoX1.5-5B (Yuan et al., 2024) | 82.17 | 82.78 | 79.76 | 96.87 | 97.35 | 98.88 | 98.31 | 50.93 | 62.79 | 65.02 |
| Omni-Video (Tan et al., 2025) | 83.00 | 84.27 | 77.92 | 98.39 | 97.68 | 99.87 | 99.10 | 56.67 | 62.48 | 64.56 |
| **UniVid (Ours)** | **85.27** | **86.44** | **80.58** | **98.96** | 97.76 | **99.88** | **99.25** | 61.83 | 64.21 | **73.03** |

| Method | Semantic Fidelity | | | | | | | | |
|---|---|---|---|---|---|---|---|---|---|
| | Object↑ | Multi-Obj↑ | Action↑ | Color↑ | Spatial↑ | Scene↑ | Appearance↑ | Temporal↑ | Overall↑ |
| EasyAnimateV5.1 (Fu et al., 2024b) | 89.57 | 66.85 | 95.60 | 77.86 | 76.11 | 54.31 | 23.06 | 24.61 | 26.47 |
| MiniMax-Video-01 (MiniMax, 2024) | **97.83** | 76.04 | 92.40 | 90.36 | 75.50 | 50.68 | 20.06 | 25.63 | 27.10 |
| Kling 1.6 (Technology, 2025) | 93.34 | 73.99 | 96.20 | 81.26 | 79.08 | 55.57 | 20.75 | 24.51 | 26.04 |
| Wan2.1-T2V-1.3B (Wang et al., 2025) | 88.81 | 74.83 | 94.00 | 82.00 | 73.04 | 41.96 | 21.81 | 23.13 | 25.50 |
| Wan2.2-TI2V-5B (Wang et al., 2025) | 89.21 | 75.23 | 94.09 | 82.43 | 72.90 | 42.36 | 21.89 | 23.78 | 26.03 |
| HunyuanVideo (Kong et al., 2024) | 86.10 | 71.66 | 93.42 | 91.60 | 68.09 | 53.69 | 19.80 | 23.89 | 26.44 |
| Gen-3 (Runway, 2024) | 87.81 | 53.64 | 96.40 | 80.90 | 65.03 | 54.57 | 24.31 | 24.71 | 26.69 |
| Vchitect-2.0 (VEnhancer) (Fan et al., 2025) | 86.61 | 68.84 | **97.20** | 87.04 | 57.55 | **56.57** | 23.73 | 25.01 | 27.57 |
| CogVideoX1.5-5B (Yuan et al., 2024) | 87.47 | 69.65 | 97.20 | 87.55 | 80.25 | 52.91 | **24.89** | 25.19 | 27.30 |
| Omni-Video (Tan et al., 2025) | 93.54 | 71.06 | 93.60 | 88.89 | 73.15 | 44.33 | 23.45 | 25.81 | 26.99 |
| **UniVid (Ours)** | 94.52 | **77.45** | 94.20 | **92.10** | **80.70** | 46.66 | 23.57 | **25.91** | **27.60** |

unseen frames by cosine similarity $\langle \mathbf{v}_i, \phi(s) \rangle$, which suits static questions whose evidence is sparse but distinctive. For shrink we start broad to preserve chronology, then apply a Maximal Marginal Relevance objective that balances relevance to $\phi(s)$ and pairwise dissimilarity within $W$, which suits dynamic questions where ordering, repetition, or transitions matter. In both regimes $W$ is kept in temporal order so the Actor can compare events across $[t_1 \rightarrow t_k]$ rather than hallucinate transitions.

**Value and critic signals.** The Evaluator provides a calibrated confidence that serves as a value proxy. Its scalar reward $r$ both triggers early stopping and conditions the Reflector. When $r$ is low, the Reflector returns a short declarative refinement of $s$ that encodes the suspected failure mode: missing entity, wrong time span, ambiguous referent, or occluded phase. This verbal update reshapes the retrieval distribution without touching model weights, yielding a form of policy gradient in the space of prompts. Our Pyramid Reflection procedure is summarized in Algorithm 1, and the high-level understanding pipeline is shown in Fig. 8. The theoretical details of Pyramid Reflection as test-time RL are provided in Appendix A.5.

The design achieves efficiency by caching frame embeddings once and reducing exploration to lightweight index updates, while the Actor reasons over compact, temporally ordered evidence with fixed global context to maintain scene priors under tight token budgets. The adaptive routing between expansion and MMR-based shrinking aligns retrieval strategies with question structure, enabling effective temporal reasoning at low computational cost.

Nevertheless, this efficiency-oriented retrieval scheme inherently operates on a sparse temporal subset rather than the full dense sequence. As a result, its ability to infer subtle motion cues, fine-grained temporal continuity, or high-frequency dynamics may be limited compared to methods that process all frames end-to-end. These dense approaches often provide more precise motion understanding and object interaction modeling, particularly in tasks where small spatial shifts or rapid temporal transitions are critical for accurate reasoning.

## 4 EXPERIMENTS

### 4.1 DATASET AND METRICS

**Datasets.** We evaluate UniVid on established benchmarks for both video generation and understanding. For generation, we train on curated samples from OpenVid-1M, a large-scale text-to-video dataset, and evaluate on VBench, a comprehensive benchmark suite for video generative models that provides fine-grained evaluation metrics across multiple dimensions. For understanding, we train on 20k samples from the ActivityNet-QA train dataset (Yu et al., 2018) and evaluate on four comprehensive video QA benchmarks: MSVD-QA (Piergiovanni et al., 2022) with 1,970 video clips and 50.5K QA pairs, MSRVTT-QA (Piergiovanni et al., 2022) with 10K videos, 243K QA pairs, TGIF-QA (Jang et al., 2017) containing 165K QA pairs for animated GIFs, and the ActivityNet-QA test

Table 2: Comparison on four video QA benchmarks (Piergiovanni et al., 2022; Jang et al., 2017; Yu et al., 2018).

| | | | Video QA Performance | | | | | | |
|---|---|---|---|---|---|---|---|---|---|
| | | **MSVD-QA** | | **MSRVTT-QA** | | **TGIF-QA** | | **ActivityNet-QA** | |
| **Method** | **LLM size** | Acc↑ | Score↑ | Acc↑ | Score↑ | Acc↑ | Score↑ | Acc↑ | Score↑ |
| FrozenBiLM (Yang et al., 2022) | 1B | 32.2 | – | 16.8 | – | 41.0 | – | 24.7 | – |
| VideoChat (Li et al., 2023) | 7B | 56.3 | 2.8 | 45.0 | 2.5 | 34.4 | 2.3 | – | 2.2 |
| LLaMA-Adapter (Zhang et al., 2023b) | 7B | 54.9 | 3.1 | 43.8 | 2.7 | – | – | 34.2 | 2.7 |
| Video-LLAMA (Zhang et al., 2023a) | 7B | 51.6 | 2.5 | 29.6 | 1.8 | – | – | 12.4 | 1.1 |
| Video-ChatGPT (Maaz et al., 2024) | 7B | 64.9 | 3.3 | 49.3 | 2.8 | 51.4 | 3.0 | 35.2 | 2.7 |
| Chat-UniVi (Jin et al., 2024) | 7B | 65.0 | 3.6 | 54.6 | 3.1 | 60.3 | 3.4 | 45.8 | 3.2 |
| Video-LLaVA (Lin et al., 2024) | 7B | 70.7 | 3.9 | 59.2 | 3.5 | 70.0 | 4.0 | 45.3 | 3.3 |
| BT-Adapter (Liu et al., 2024) | 7B | 67.5 | 3.7 | 57.0 | 3.2 | – | – | 45.7 | 3.2 |
| Valley-v3 (Luo et al., 2023) | 7B | 60.5 | 3.3 | 51.1 | 2.9 | – | – | 45.1 | 3.2 |
| FreeVA (Wu, 2024) | 7B | 73.8 | 4.1 | 60.0 | 3.5 | – | – | 51.2 | 3.5 |
| DeepStack-L (Meng et al., 2024) | 7B | 76.0 | 4.0 | – | – | – | – | 49.3 | 3.1 |
| IG-VLM (LLaVA-v1.6) (Kim et al., 2024) | 7B | 78.8 | 4.1 | 63.7 | 3.5 | – | 4.0 | 54.3 | 3.4 |
| SF-LLaVA-7B (Xu et al., 2024) | 7B | 79.1 | 4.1 | **65.8** | **3.6** | **78.7** | **4.2** | 55.5 | 3.4 |
| **UniVid (Ours)** | 7B | **80.1** | **4.2** | 61.4 | 3.4 | 75.0 | 4.1 | **58.8** | **3.6** |

dataset (Yu et al., 2018) with 58,000 QA pairs on 5,800 complex web videos. These datasets cover diverse temporal reasoning scenarios across short to medium-length video clips, ranging from brief animated sequences to multi-minute activity videos.

**Evaluation metrics.** For video generation, we evaluate on VBench across multiple fine-grained dimensions: Technical Quality metrics including Subject consistency, Background preservation, Temporal flickering, Motion smoothness, and Dynamic degree; Aesthetic Quality measures covering overall visual appeal and imaging quality; and Semantic Fidelity metrics assessing Object accuracy, Multi-object handling, Action fidelity, Color accuracy, Spatial relationships, Scene consistency, Appearance preservation, and Temporal coherence. For video understanding, we report average accuracy and scores on each benchmark dataset.

## 4.2 IMPLEMENTATION DETAILS

We adopt a three-stage hierarchical training recipe. It initializes UniVid from strong public checkpoints to reduce compute. For generation, we couple the BAGEL-7B (Deng et al., 2025) with Wan 2.2 5B TI2V model (Wang et al., 2025) via a textual adapter and LoRA on DiT (Peebles & Xie, 2023), keeping other weights frozen. For understanding, we tune only the connector and the last two ViT blocks on ActivityNet QA (Yu et al., 2018) with dialog style supervision while the LLM remains frozen. Finally, we co-train both tasks to refine the connector and obtain additive gains. Sequence parallelism enables long high-resolution clips. For details, see Appendix A.2.

For generation, we use a flow-matching ODE sampler with classifier-free guidance and a universal negative prompt. Unless noted, videos are sampled at $1280 \times 704$ resolution, 121 frames at 24 fps; the guidance scale is set to 5.0 for both T2V and I2V with 50 inference steps. At input time, the LLM receives the text prompt together with image ViT embeddings and VAE latents; it outputs conditional textual tokens. During generation, Wan 2.2 consumes these conditional textual tokens and image via cross-attention. Our Temperature Modality Alignment schedule applies a cosine-scheduled text gain that transitions from $\alpha_{\text{txt}} = 1.3$ to 1.0 over the first 40% of denoising steps ($u \in [0, 0.4]$), then maintains $\alpha_{\text{txt}} = 1.0$ for the remaining steps. This enhances text guidance during early denoising when structural decisions are made, while allowing finer details to emerge in later stages.

For understanding, we uniformly sample a pool of $N = 64$ frames per video and cache their SigLIP2 image embeddings; subsequent selection reuses cached features. Global context is a caption summarized from 16 uniformly spaced seed frames. Query–image ranking uses SigLIP2 cosine similarity with L2-normalized features and batch size 64. Static questions follow a $4 \rightarrow 8 \rightarrow 16$ keyframe schedule. Dynamic questions follow $64 \rightarrow 32 \rightarrow 16$ with MMR down-selection, $\lambda = 0.5$. Confidence is accepted when the Evaluator's score is at least $0.7$ or the verdict is accept, with at most $R \leq 3$ rounds. The LLM determines routing between static and dynamic modes. For implementation, we use DeepSeek v3.1 to serve as the Evaluator and determine the type of questions and Qwen-plus to serve as the Reflector. Full prompt texts are listed in the Appendix A.4.

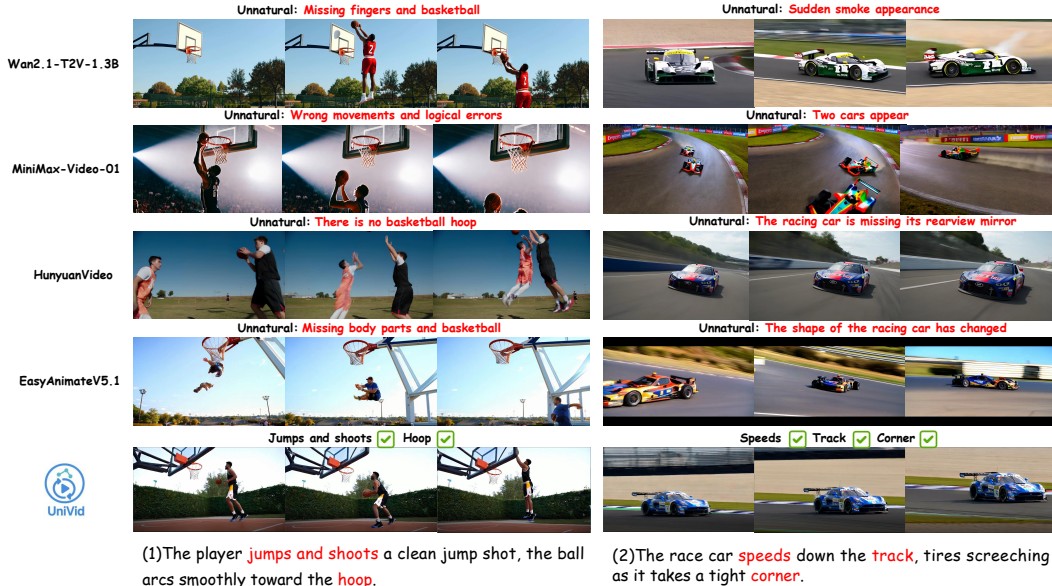

Figure 3: Comparisons with State-of-the-Art Video Generation Models (Wang et al., 2025; Mini-Max, 2024; Kong et al., 2024; Fu et al., 2024b).

## 4.3 MAIN RESULTS

**Generation quantitative results.** We evaluate UniVid on the challenging VBench-Long benchmark (Huang et al., 2024). As shown in Tab. 1, UniVid establishes a new state of the art with an overall score of 85.27, outperforming prior leading systems such as EasyAnimateV5.1 (Fu et al., 2024b), MiniMax-Video-01 (MiniMax, 2024), and Kling 1.6 (Technology, 2025). In particular, UniVid exhibits clear advantages in semantic alignment (80.58), highlighting its superior capability in faithfully rendering objects, actions, and multi-object interactions. On the technical side, it attains near-perfect temporal (99.88) and motion (99.25) consistency, validating the effectiveness of our long-context dynamics module. Moreover, UniVid delivers the best imaging score (73.03), reflecting sharper details and more stable visual quality compared with prior systems, as shown in Fig. 1, which demonstrates high-quality visual generation.

Beyond overall scores, UniVid demonstrates consistent gains in semantic fidelity. As summarized in the Semantic Fidelity block of Tab. 1, it achieves leading results on multi-object reasoning (77.45), color faithfulness (92.10), and spatial grounding (80.70), while remaining competitive in action depiction and appearance consistency. These improvements suggest that our design choices—particularly the integration of hierarchical scene representation with dynamic frame alignment—substantially enhance both controllability and alignment with textual prompts. Taken together, the results indicate that UniVid pushes forward the frontier of long-horizon text-to-video generation by simultaneously ensuring high-fidelity semantics and strong technical as well as aesthetic quality. More examples of video generation can be seen in Appendix A.3.

**Generation qualitative results.** Fig. 3 compares UniVid with Wan2.1-T2V-1.3B (Wang et al., 2025), MiniMax-Video-01 (MiniMax, 2024), HunyuanVideo (Kong et al., 2024), and EasyAnimateV5.1 (Fu et al., 2024b). Competing models often show missing basketballs or distorted cars, while UniVid generates coherent jump shots and realistic racing scenes with stable dynamics and faithful semantics.

**Understanding quantitative evaluation.** Across MSVD-QA (Piergiovanni et al., 2022), MSRVTT-QA (Piergiovanni et al., 2022), TGIF-QA (Jang et al., 2017), and ActivityNet-QA (Yu et al., 2018), UniVid sets the 7B-scale state of the art on MSVD-QA and ActivityNet-QA and remains competitive on the other two (Tab. 2), despite a smaller post-training set and no test-time ensembling. Joint finetuning of generation and understanding with Pyramid Reflection strengthens the abilities these datasets emphasize: better action–entity binding and object or attribute grounding

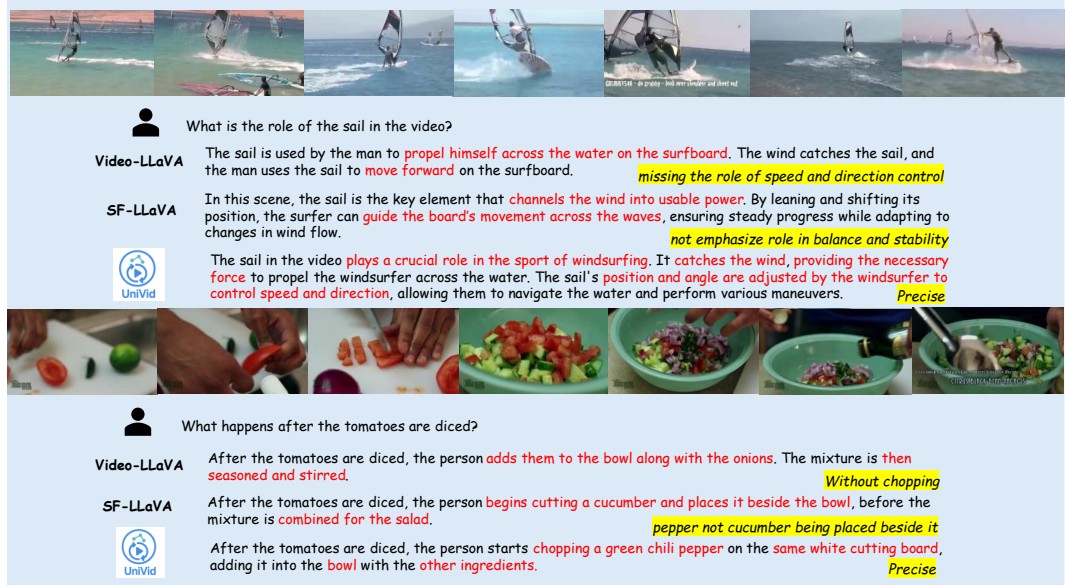

Figure 4: Comparisons of State-of-the-Art Video Understanding Models (Lin et al., 2024; Xu et al., 2024).

in short open-domain clips, stronger temporal reasoning over frame sequences, and more reliable long-range evidence retrieval in untrimmed videos.

As illustrated before, UniVid performs robust multi-frame reasoning with our Pyramid Reflection loop. Starting from a global caption and automatic type detection, the system first produces an initial answer, which is then scored by the evaluator; when evidence is insufficient, the reflector issues a refined, declarative query that re-ranks keyframes toward the true scene. This Pyramid Reflection steers attention from opening credits to the lane shots, yielding a consistent interpretation of roles (in the example of Fig. 8: bowler and nearby teammate/coach) grounded in the visual context rather than spurious cues. The dynamic keyframe schedule reduces the number of inspected frames while maintaining accuracy, demonstrating both evidence tracing and efficiency gains in short-clip understanding. More examples of video understanding can be seen in Appendix A.3.

**Understanding qualitative results.** We compare UniVid with Video-LLaVA (Lin et al., 2024) and SF-LLaVA (Xu et al., 2024) on video QA; as shown in Fig. 4, baselines often give plausible but incomplete statements. These examples highlight UniVid's stronger action–entity binding, temporal reasoning, and resistance to distractor frames, yielding precise and concise answers. Additionally, we conduct systematic ablation experiments to validate the contributions of UniVid. The results and analyses are provided in the Appendix A.6.

## 5 CONCLUSION

We introduced UniVid, a unified video model that couples an MLLM with a diffusion decoder via a lightweight conditioning adapter to both understand and generate videos. Two key mechanisms enable this: Temperature Modality Alignment schedules cross-modal attention across flow steps to preserve prompt faithfulness while refining details, and Pyramid Reflection performs query-driven keyframe selection for efficient temporal reasoning. With these components, UniVid achieves state-of-the-art or competitive results on VBench-Long and multiple video-QA benchmarks while avoiding costly retraining of image-centric backbones. We release UniVid to support research on practical, controllable, and truly unified video intelligence.

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

# A APPENDIX

## A.1 LLM USE DECLARATION

Large Language Models (ChatGPT) were used exclusively to improve the clarity and fluency of English writing. They were not involved in research ideation, experimental design, data analysis, or interpretation. The authors take full responsibility for all content.

## A.2 HIERARCHICAL POST TRAINING

**Initialization.** To avoid the prohibitive cost of training a unified video model from scratch, we bootstrap UniVid from strong, publicly available checkpoints and finetune only small subsets of parameters. Our architecture follows the BAGEL (Deng et al., 2025) design framework, adopting its multimodal integration approach with three key components: Qwen2 (Yang et al., 2024) as the LLM backbone with standard architectural choices such as RMSNorm (Zhang & Sennrich, 2019), SwiGLU (Shazeer, 2020), RoPE (Su et al., 2024), GQA (Ainslie et al., 2023), and QK-Norm (Henry et al., 2020) for training stability, SigLIP2-so400m/14 (Tschannen et al., 2025) as the ViT (Dosovitskiy et al., 2021) encoder for visual understanding with NaViT support for native aspect ratios, and a pre-trained FLUX VAE with 8× downsampling and frozen weights. The framework interleaves text, ViT, and VAE tokens within the LLM using generalized causal attention, where tokens attend to all preceding modality splits while maintaining appropriate attention patterns within each modality.

**Data curation and formatting.** For understanding, we align our data format with the dialog style used by Video-ChatGPT (Maaz et al., 2024). ActivityNet-QA annotations $(\texttt{video\_id}, q, a)$ are converted into structured conversations. Specifically, each sample is represented as a JSON object containing three fields: (1) an identifier, (2) a video reference, and (3) a conversations array consisting of two turns, a user query and the corresponding model response. For generation, we curate a subset of OpenVid-1M to form text/image to video pairs. Videos are uniformly sub-sampled and preprocessed identically to inference.

**Stage I generation branch alignment.** We couple the MLLM with Wan 2.2 and adapt the conditioning path so that MLLM-produced tokens can reliably steer synthesis. Concretely, we (i) insert a textual adapter between the LLM tokens, with dynamic sequence length adaptation, and (ii) apply LoRA to the DiT cross-attention layers; all other DiT/MLLM weights remain frozen. Training uses a standard flow-matching objective with classifier-free guidance dropout on text, optimizing only the context projector and LoRA parameters. This stage preserves MLLM's native understanding while aligning Wan's generation to the rich semantics emitted by MLLM.

**Stage II understanding adaptation.** We finetune for video QA on ActivityNet-QA using 20k samples from the dataset. Each sample concatenates the question with a `<video>` placeholder, and we feed a multi-frame clip obtained by uniform sampling. Frames are encoded by the ViT into visual tokens and projected to the LLM space via the connector. We adopt instruction SFT for video: compute autoregressive cross-entropy only on the assistant turns; user tokens are fully masked to prevent label leakage. To keep compute moderate while injecting temporal cues, we finetune only the last two ViT blocks and the connector for 4 epochs, keeping the LLM frozen.

**Stage III joint training.** Finally, we co-train generation and understanding to let the two branches benefit from each other. During joint training, we gradually activate the VAE feature branch and alternate which modules are trainable. In the initial phase, only the last few ViT layers and connector are trainable and the adapter is frozen, so the MLLM is forced to encode features in ViT that are beneficial for both understanding and generation. Next, we freeze the ViT and progressively introduce VAE features into the adapter input, enabling the DiT LoRA to learn how to generate guided by different mixing ratios of ViT and VAE features. Finally, we co-train both the ViT layers, connector, adapter and DiT to fully exploit the complementary strengths of semantic ViT features and detail-rich VAE features. On the understanding side, Fig. 10 shows that joint training leads to faster convergence and consistently higher MSVD-QA (Piergiovanni et al., 2022) accuracy than single-task training. On the generation side, Stage III further improves the VBench-Long overall score

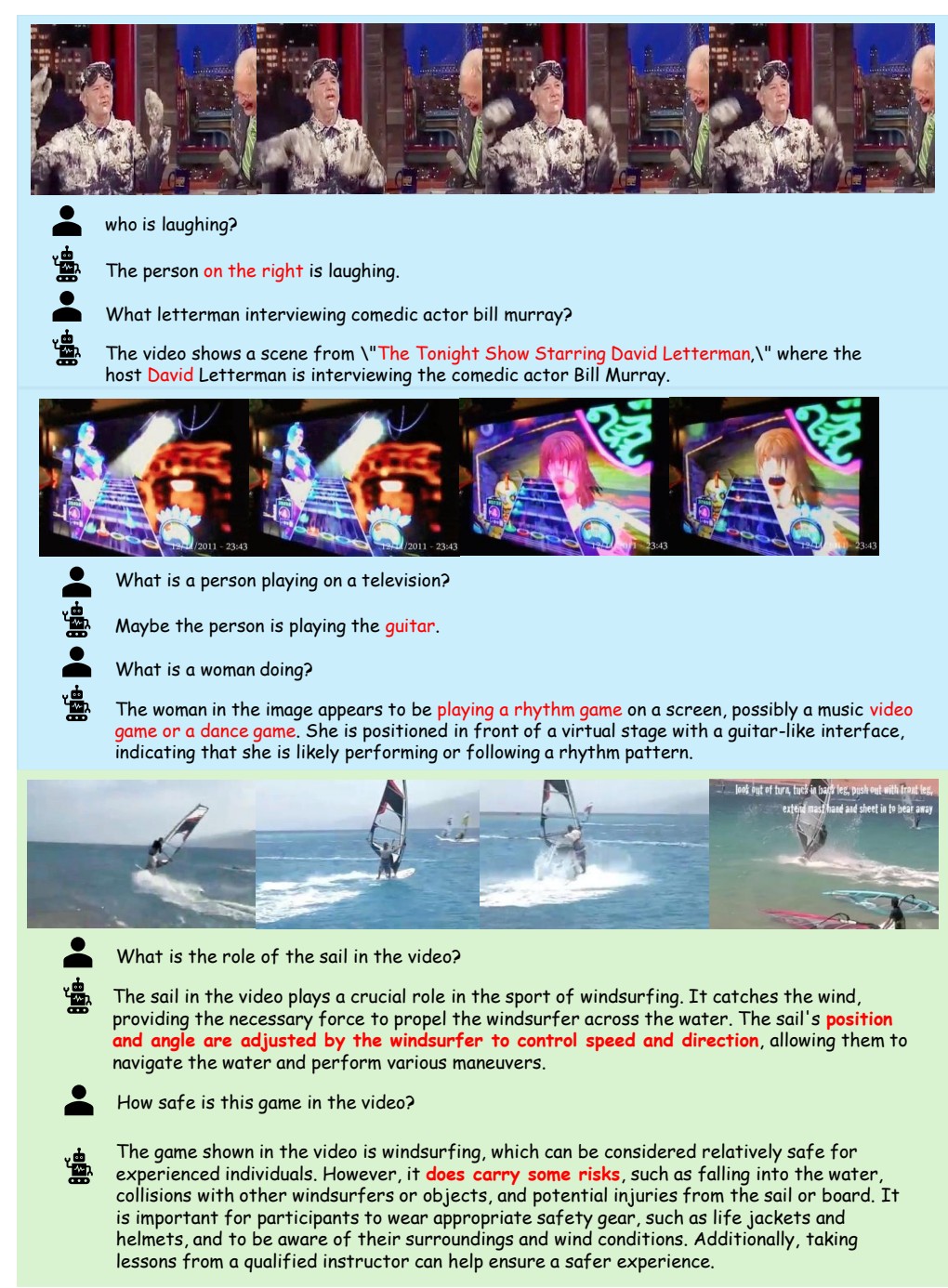

Figure 5: The qualitative results of the video understanding. Blue for static questions, green for dynamic questions.

from 79.28 to 85.27 and boosts most technical and semantic dimensions, as summarized in Tab. 6, confirming that better video understanding feedback translates into higher-quality video generation.

A.3 MORE EXAMPLES OF VIDEO GENERATION AND UNDERSTANDING.

We provide more examples of video understanding and generation in Fig. 5 and Fig. 6

**Text to Video**

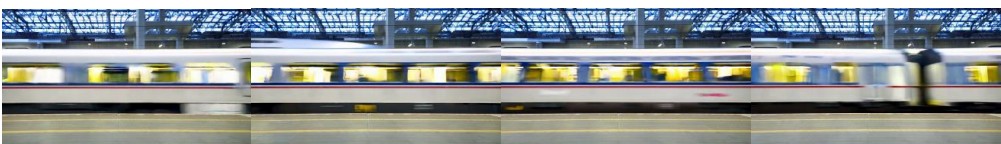

A dolphin leaps out of the ocean, splashing water as it dives back in.

Two anthropomorphic cats in comfy boxing gear and bright gloves fight intensely on a spotlighted stage.

A futuristic drone weaves quickly between skyscrapers, lights glowing in the night sky.

A high-speed train rushes past the station, its motion blurring in the background.

**Text and Image to Video**

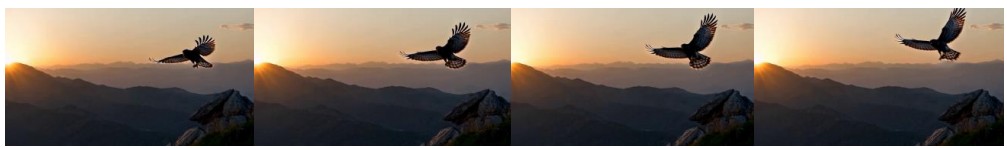

A hawk soars above the mountains, wings spread wide against the sunset.
(from image)

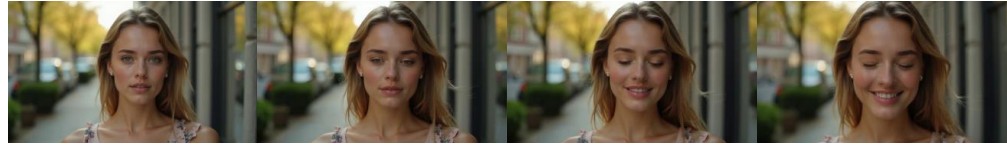

A cinematic video of a young woman with natural makeup and long blonde hair, standing on a sunlit street with blurred trees and cars in the background. The camera slowly moves closer as her hair gently flows with the breeze. She softly smiles and blinks, creating a natural and elegant moment. Warm golden hour lighting, realistic style, high detail, 4K.
(from image)

Figure 6: The qualitative results of T2V and TI2V generation.

## A.4 TEXT PROMPTS USED IN THE UNDERSTANDING

**Role.** Classify a video question as `static` or `dynamic`. Output JSON only.
**Definitions.**
- `dynamic`: requires temporal reasoning such as counting, repetition, order, or changes over time (e.g., "how many times", "before/after", "first/last").
- `static`: can be answered from a small set of unordered frames (identity, attribute, location, scene, one-shot action).

**Question.** {*question*}
**Return.** Single-line JSON with fields: `qtype` (`"static"` or `"dynamic"`), `rationale` (1–2 short phrases; no extra text).

### 1: Question Type Classification Prompt

**Role.** Summarize chronologically ordered frame notes into a compact global caption. Do not invent facts.
**Input.** Frame-wise notes (earlier → later):
- {*note_1*}
- {*note_2*}
. . .
**Write.** One global caption (2–4 sentences) that connects multiple frames, focusing on: (1) moving entities with consistent appearance and actions across time; (2) static scene objects and their states; (3) temporal hints only if explicitly evidenced (e.g., "then", "later", "repeatedly"). Style: terse and factual; no bullet lists, storytelling, or frame-by-frame recitation.

### 2: Frame Summarization Prompt

**Role.** Precise evaluator for video-QA. Return a *single-line* JSON only (no Markdown/code).
**Keys.** `score` (float 0..1), `verdict` (`"accept"` if score $\geq$ 0.7 else `"reject"`), `brief_reason` (1–2 short bullets).
**Example user.** {*one_shot_user*}
**Example assistant.** {*one_shot_assistant*}
**Your task.** Given the current case, output the JSON only.

### 3: Answer Evaluation Prompt

**Role.** Reflector in a video-understanding pipeline. You receive the question, a global caption (from 16 uniformly sampled frames), the last answer (low confidence/rejected), and its evaluation JSON.
**Objective.** Analyze why the answer likely fails (missing object, wrong span, ambiguity, etc.) and produce a single short *declarative* retrieval text for the next round of keyframe selection.
**Strict rules.** (1) Output JSON only with key `refined_query`. (2) `refined_query` $\leq$ 25 tokens, declarative statement (not a question), capturing disambiguating cues (entities, attributes, actions, temporal hints, viewpoint). (3) If confidence is already good (score $\geq$ 0.7 or verdict=`"accept"`), return an empty string. (4) Prefer concrete visual cues (colors, clothing, object names, motion phase, timestamps, left/right, first/last). (5) No speculation or unseen entities.
**Inputs.** Question: {*question*}  Global caption: {*global_caption*}  Last answer: {*last_answer*}  Evaluation JSON: {*eval_json*}
**Return.** {`"refined_query"`: `"..."`}

### 4: Reflection Prompt

Table 3: Ablation study of UniVid on VBench-Long. *w/o* means "without". Best results are **bold**.

| Model | Overall Scores | | | Technical Quality | | | | | Aesthetic Quality | |
|---|---|---|---|---|---|---|---|---|---|---|
| | Total Score↑ | Quality↑ | Semantic↑ | Subject↑ | Background↑ | Temporal↑ | Motion↑ | Dynamic↑ | Aesthetic↑ | Imaging↑ |
| UniVid (base) | 76.25 | 77.11 | 72.82 | 93.82 | 93.43 | 94.15 | 94.04 | 57.16 | 58.47 | 65.65 |
| UniVid (w/o MLLM) | 77.82 | 78.69 | 74.32 | 94.55 | 94.78 | 95.19 | 94.79 | 58.08 | 59.88 | 66.01 |
| UniVid (w/o TMA) | 80.42 | 81.51 | 76.04 | 96.55 | 95.91 | 97.12 | 96.25 | 59.98 | 62.08 | 67.10 |
| **UniVid (Full)** | **85.27** | **86.44** | **80.58** | **98.96** | **97.76** | **99.88** | **99.25** | **61.83** | **64.21** | **73.03** |

| Model | Semantic Fidelity | | | | | | | | |
|---|---|---|---|---|---|---|---|---|---|
| | Object↑ | Multi-Obj↑ | Action↑ | Color↑ | Spatial↑ | Scene↑ | Appearance↑ | Temporal↑ | Overall↑ |
| UniVid (base) | 89.53 | 73.32 | 89.41 | 87.86 | 76.13 | 42.32 | 19.03 | 21.60 | 22.48 |
| UniVid (w/o MLLM) | 90.80 | 74.37 | 90.12 | 87.99 | 76.63 | 43.32 | 20.57 | 22.26 | 22.98 |
| UniVid (w/o TMA) | 91.51 | 75.42 | 91.53 | 89.33 | 77.58 | 44.61 | 21.03 | 23.62 | 24.13 |
| **UniVid (Full)** | **94.52** | **77.45** | **94.20** | **92.10** | **80.70** | **46.66** | **23.57** | **25.91** | **27.60** |

Table 4: Ablation study on TMA schedules on VBench-Long. *w/o* means "without". Best results are **bold**.

| Model | Overall Scores | | | Technical Quality | | | | | Aesthetic Quality | |
|---|---|---|---|---|---|---|---|---|---|---|
| | Total Score↑ | Quality↑ | Semantic↑ | Subject↑ | Background↑ | Temporal↑ | Motion↑ | Dynamic↑ | Aesthetic↑ | Imaging↑ |
| UniVid (w/o TMA) | 80.42 | 81.51 | 76.04 | 96.55 | 95.91 | 97.12 | 96.25 | 59.98 | 62.08 | 67.10 |
| UniVid (Constant) | 82.72 | 83.96 | 77.78 | 97.81 | 96.41 | 98.12 | 98.01 | 60.11 | 63.47 | 70.65 |
| UniVid (Step) | 82.80 | 84.35 | 76.59 | 97.32 | 96.74 | 98.15 | 98.54 | 59.71 | 63.91 | 71.19 |
| UniVid (Linear) | 83.30 | 84.51 | 78.47 | 97.45 | 96.78 | 98.20 | 98.76 | 60.01 | 63.88 | 71.01 |
| **UniVid (Consine)** | **85.27** | **86.44** | **80.58** | **98.96** | **97.76** | **99.88** | **99.25** | **61.83** | **64.21** | **73.03** |

| Model | Semantic Fidelity | | | | | | | | |
|---|---|---|---|---|---|---|---|---|---|
| | Object↑ | Multi-Obj↑ | Action↑ | Color↑ | Spatial↑ | Scene↑ | Appearance↑ | Temporal↑ | Overall↑ |
| UniVid (w/o TMA) | 91.51 | 75.42 | 91.53 | 89.33 | 77.58 | 44.61 | 21.03 | 23.62 | 24.13 |
| UniVid (Constant) | 92.52 | 76.81 | 92.40 | 90.81 | 79.13 | 45.29 | 22.01 | 24.19 | 25.41 |
| UniVid (Step) | 91.78 | 75.81 | 91.41 | 89.88 | 78.13 | 44.89 | 21.78 | 23.54 | 24.31 |
| UniVid (Linear) | 92.80 | 76.32 | 92.11 | 90.98 | 79.61 | 45.25 | 22.56 | 24.21 | 26.91 |
| **UniVid (Consine)** | **94.52** | **77.45** | **94.20** | **92.10** | **80.70** | **46.66** | **23.57** | **25.91** | **27.60** |

---

**Role.** Assist video understanding via per-frame analysis. Describe the main objects and actions in *this single frame* concisely.
**Focus.** (1) Living entities: distinct entities (appearance, clothing, color, species), likely roles, and what each is doing (verb phrases). (2) Static objects & scene: salient items and states (color, shape, on/off, open/closed, broken/intact), plus scene context (indoor/outdoor, location hints).
**Style.** Specific but brief; no speculation; 2–4 short sentences.

### 5: Single-Frame Analysis Prompt

---

**Role.** Answer concisely using only the question and the global video caption.
**Inputs.** Question: {*question*}     Global caption (may miss fine details): {*global_caption*}
**Instruction.** Produce one short answer (1–2 sentences). If information is insufficient, reply: *"Not enough evidence from global caption."*

### 6: Global Answer Prompt

## A.5  Pyramid Reflection as Test-time RL

We cast Pyramid Reflection as a test-time reinforcement learning procedure operating on an ordered evidence set. At round $r$, the state is $x_r = (s_r, W_r, C_g)$, where $s_r$ is a short search text, $W_r$ is the ordered working set of frames, and $C_g$ is a global caption distilled once from uniformly sampled seeds. The action reconfigures $W_r$ given $s_r$ via an expand or shrink policy. The Actor answers from $(W_r, C_g)$, and the Evaluator returns a score $R_r \in [0, 1]$ and a verdict that controls early stopping. All frame embeddings are computed once and cached; later rounds update indices and similarity or diversity scores only.

Table 5: Ablation study of the generation branch of UniVid to verify the effectiveness of encoder setting. *w/o* means "without". Best results are **bold**.

| Model | Overall Scores | | | Technical Quality | | | | | Aesthetic Quality | |
|---|---|---|---|---|---|---|---|---|---|---|
| | Total Score↑ | Quality↑ | Semantic↑ | Subject↑ | Background↑ | Temporal↑ | Motion↑ | Dynamic↑ | Aesthetic↑ | Imaging↑ |
| UniVid (w/o ViT) | 48.53 | 57.16 | 46.37 | 74.51 | 72.91 | 74.02 | 74.23 | 46.91 | 47.01 | 55.10 |
| UniVid (w/o VAE) | 71.78 | 71.90 | 71.75 | 89.43 | 88.75 | 90.19 | 89.80 | 57.23 | 58.86 | 67.12 |
| **UniVid (Ours, VAE & ViT Encoder)** | **85.27** | **86.44** | **80.58** | **98.96** | **97.76** | **99.88** | **99.25** | **61.83** | **64.21** | **73.03** |

| Model | Semantic Fidelity | | | | | | | | |
|---|---|---|---|---|---|---|---|---|---|
| | Object↑ | Multi-Obj↑ | Action↑ | Color↑ | Spatial↑ | Scene↑ | Appearance↑ | Temporal↑ | Overall↑ |
| UniVid (w/o ViT) | 72.41 | 54.41 | 75.51 | 74.31 | 58.68 | 32.69 | 14.12 | 15.63 | 17.15 |
| UniVid (w/o VAE) | 87.23 | 69.54 | 87.34 | 88.92 | 74.32 | 39.27 | 20.54 | 21.61 | 22.12 |
| **UniVid (Ours, VAE & ViT Encoder)** | **94.52** | **77.45** | **94.20** | **92.10** | **80.70** | **46.66** | **23.57** | **25.91** | **27.60** |

Table 6: Stage I vs Stage III performance on VBench-Long to verify the effect of hierarchical joint training on video generation. *w/o* means "without". Best results are **bold**.

| Model | Overall Scores | | | Technical Quality | | | | | Aesthetic Quality | |
|---|---|---|---|---|---|---|---|---|---|---|
| | Total Score↑ | Quality↑ | Semantic↑ | Subject↑ | Background↑ | Temporal↑ | Motion↑ | Dynamic↑ | Aesthetic↑ | Imaging↑ |
| UniVid (Satge I) | 79.28 | 80.38 | 74.90 | 94.23 | 94.19 | 95.31 | 96.32 | 58.98 | 61.91 | 70.11 |
| **UniVid (Joint, Stage III)** | **85.27** | **86.44** | **80.58** | **98.96** | **97.76** | **99.88** | **99.25** | **61.83** | **64.21** | **73.03** |

| Model | Semantic Fidelity | | | | | | | | |
|---|---|---|---|---|---|---|---|---|---|
| | Object↑ | Multi-Obj↑ | Action↑ | Color↑ | Spatial↑ | Scene↑ | Appearance↑ | Temporal↑ | Overall↑ |
| UniVid (Stage I) | 90.12 | 75.59 | 90.98 | 89.91 | 77.52 | 44.57 | 20.51 | 21.12 | 24.01 |
| **UniVid (Joint, Stage III)** | **94.52** | **77.45** | **94.20** | **92.10** | **80.70** | **46.66** | **23.57** | **25.91** | **27.60** |

Frame selection uses a vision–language retriever with cosine similarity. Let $\phi(s)$ be the text embedding and $\{\mathbf{v}_i\}_{i=1}^N$ the cached frame embeddings:

$$\text{sim}(i, s) = \langle \widehat{\mathbf{v}}_i, \widehat{\phi(s)} \rangle. \tag{6}$$

We define a soft retrieval policy over the pool $P$:

$$\pi(i \mid s) = \frac{\exp\big(\text{sim}(i, s)/\tau\big)}{\sum_{j \in P} \exp\big(\text{sim}(j, s)/\tau\big)}. \tag{7}$$

Sampling sequentially without replacement with joint probability $\prod_{\ell=1}^K \pi(i_\ell \mid s, i_{<\ell})$ and respecting chronology yields $W_s$.

In the expand mode, at target size $K_t$ we add the top $m$ unseen frames by similarity (no duplicates):

$$\Delta_t = \arg\max_{i \in P \setminus S_{\text{sel}}}^m \text{sim}(i, s_{t-1}), \qquad S_{\text{sel}} \leftarrow S_{\text{sel}} \cup \Delta_t, \quad m = K_t - |S_{\text{sel}}|. \tag{8}$$

In the shrink mode, with current $S_{\text{sel}}$ and target $K_t \in \{32, 16\}$, we apply Maximal Marginal Relevance:

$$S_{\text{sel}} = \arg\max_{S \subseteq S_{\text{sel}}, |S|=K_t} \sum_{i \in S} \Big[ \lambda \, \text{sim}(i, s_{t-1}) - (1-\lambda) \max_{j \in S \setminus \{i\}} \text{sim}(i, j) \Big]. \tag{9}$$

We adopt a verbal policy–improvement view (Shinn et al., 2023). Let the objective be the expected Evaluator value under the retrieval policy:

$$J(s) = \mathbb{E}_{i_{1:K} \sim \pi(\cdot \mid s)}[V(W_s)], \tag{10}$$

with

$$V(W_s) = \mathbb{E}\big[R \mid W_s, C_g\big]. \tag{11}$$

Using the likelihood–ratio identity with a baseline $b$ yields

$$\nabla_s J(s) = \mathbb{E}\left[ \left( \sum_{t=1}^K \nabla_s \log \pi(i_t \mid s, i_{<t}) \right) (R - b) \right]. \tag{12}$$

A single ascent step motivates a verbal update to the search text:

$$s_{r+1} = s_r + \eta \Big( \sum_{t=1}^K \nabla_s \log \pi(i_t \mid s_r, i_{<t}) \Big) (R_r - b), \tag{13}$$

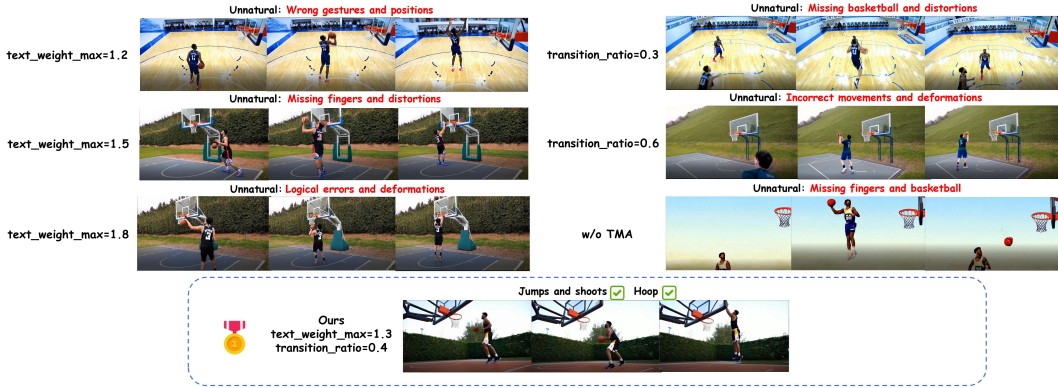

Figure 7: Ablation Study on Temperature Modality Alignment.

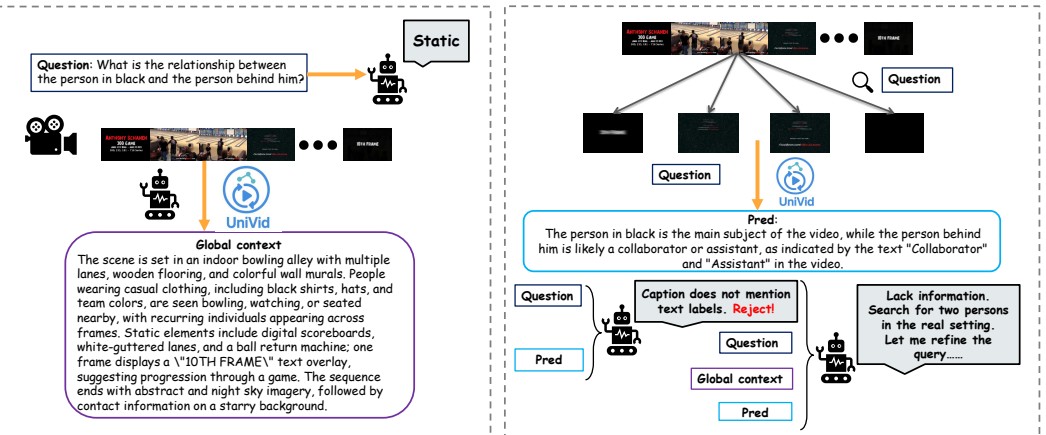

**Stage 1**: Extract the global caption and determine the question type. **Stage 2**: Initial response with subsequent evaluation and reflection.

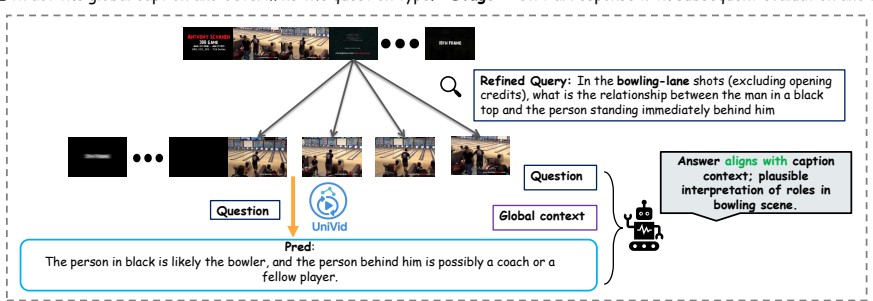

**Stage 3**: Further information augmentation with a focus on understanding and evaluation.

Figure 8: The pipeline of the video understanding.

where we use the softmax score function with $g_i(s) := \nabla_s \text{sim}(i, s)$ and $\bar{g}(s) := \mathbb{E}_{j \sim \pi(\cdot|s)} g_j(s)$:
$\nabla_s \log \pi(i \mid s) = \tau^{-1}(g_i(s) - \bar{g}(s))$, so the edit in $s$ aligns with frames that explain higher return through the text encoder $\phi(\cdot)$. Practically, the reflector inserts temporally and semantically discriminative cues (entities, colors, viewpoints, before/after, first/last, motion phase), which increases $\text{sim}(i, s)$ for diagnostic frames and decreases it for distractors, implementing Eq. 13 in language space without parameter updates.

To connect the update with both expand and shrink, we use a piecewise-smooth set surrogate that trades relevance against redundancy (subgradients at ties):

$$\tilde{V}(W_s) = \frac{1}{K} \sum_{i \in W_s} \text{sim}(i, s) \; - \; \gamma \max_{i \neq j \in W_s} \text{sim}(i, j). \tag{14}$$

| Timestep | 1s | 7s | 14s |

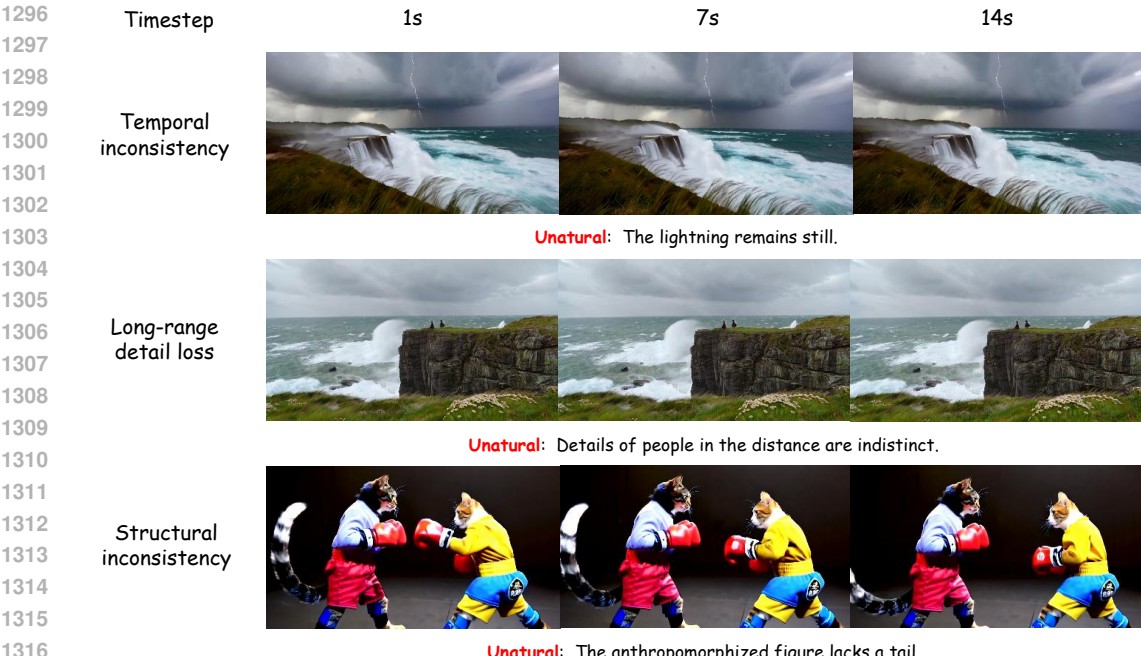

Figure 9: Categorized Failure Modes in Video Generation.

Table 7: Ablation study of UniVid on four video QA benchmarks. Acc. denotes accuracy (%), Score denotes average rating (0–5). Best results are **bold**.

| Methods | MSVD-QA | | MSRVTT-QA | | TGIF-QA | | ActivityNet-QA | |
|---|---|---|---|---|---|---|---|---|
| | Acc↑ | Score↑ | Acc↑ | Score↑ | Acc↑ | Score↑ | Acc↑ | Score↑ |
| UniVid (Base) | 64.1 | 3.3 | 48.9 | 2.8 | 54.2 | 3.0 | 39.8 | 3.0 |
| UniVid (w/o finetune) | 71.1 | 3.9 | 52.2 | 3.0 | 63.5 | 3.6 | 46.5 | 3.2 |
| UniVid (w/o Reflection) | 73.1 | 4.0 | 55.0 | 3.1 | 64.6 | 3.6 | 52.0 | 3.4 |
| UniVid (Full) | **80.1** | **4.2** | **61.4** | **3.4** | **75.0** | **4.1** | **58.8** | **3.6** |

Since $\partial \operatorname{sim}(i, s)/\partial s$ points toward $\mathbf{v}_i$ via $\phi(s)$, the gradient $\nabla_s \tilde{V}(W_s)$ is aligned with the direction in Eq. 12. If the reflector's edit correlates with the advantage $A_r = R_r - b$, then for a sufficiently small step size $\eta$ the expected first-order improvement satisfies

$$\mathbb{E}[J(s_{r+1}) - J(s_r)] \approx \eta \, \mathbb{E}\left[\left\langle \sum_t \nabla_s \log \pi(i_t \mid s_r, i_{<t}), \; s_{r+1} - s_r \right\rangle A_r\right] \geq 0. \tag{15}$$

Early stopping is triggered when the Evaluator score exceeds a fixed threshold:

$$\text{stop at round } r \text{ if } R_r \geq \tau, \qquad \tau = 0.7. \tag{16}$$

With cached features, each round requires only similarity and diversity scoring together with reasoning over a compact, temporally ordered $W_r$, which concentrates the Actor on temporal relations under a tight token budget and improves video understanding with low computational cost.

## A.6 ABLATION STUDY

**Ablation on video generation.** Tab. 3 presents an ablation on VBench-Long disentangling the roles of our two main components. Removing the multi-level language modeling module (w/o MLLM) mainly hurts the semantic-fidelity metrics that require precise spatial layout and appearance preservation, while the low-level technical quality remains relatively stable. In contrast, disabling Temperature Modality Alignment (w/o TMA) leads to a clear drop in temporal and motion-related scores, indicating that the denoising process becomes less stable over long horizons even though per-frame quality is still high. The full UniVid model consistently achieves the best performance across technical, aesthetic, and semantic dimensions, suggesting that multi-level language modeling and TMA are complementary: the former strengthens multi-object, spatial, and appearance grounding, whereas the latter enforces temporally coherent, prompt-faithful dynamics during generation.

Table 8: Ablation on Evaluator/Reflector Model Size (Hereafter, we use E to denote the Evaluator and R to denote the Reflector). Acc. denotes accuracy (%), Score denotes average rating (0–5).

| Methods | MSVD-QA | | MSRVTT-QA | | TGIF-QA | | ActivityNet-QA | |
|---|---|---|---|---|---|---|---|---|
| | Acc↑ | Score↑ | Acc↑ | Score↑ | Acc↑ | Score↑ | Acc↑ | Score↑ |
| UniVid (Qwen2-7B E&R ) | 76.9 | 3.9 | 57.4 | 3.2 | 71.8 | 3.9 | 56.7 | 3.5 |
| UniVid (LLaMA-3 8B E and LLaVA-1.6 7B R) | 78.2 | 4.0 | 59.1 | 3.3 | 72.4 | 4.0 | 56.8 | 3.5 |
| UniVid (Qwen2-7B R) | 78.5 | 4.0 | 59.0 | 3.3 | 71.8 | 3.9 | 57.6 | 3.5 |
| UniVid (Qwen2-7B E) | 77.4 | 3.9 | 58.4 | 3.2 | 72.2 | 3.9 | 57.3 | 3.5 |
| UniVid (Ours) | 80.1 | 4.2 | 61.4 | 3.4 | 75.0 | 4.1 | 58.8 | 3.6 |

Table 9: Ablation study of the understanding branch of UniVid to verify the effectiveness of encoder setting. Acc. denotes accuracy (%), Score denotes average rating (0–5). *w/o* means "without". Best results are **bold**.

| Methods | MSVD-QA | | MSRVTT-QA | | TGIF-QA | | ActivityNet-QA | |
|---|---|---|---|---|---|---|---|---|
| | Acc↑ | Score↑ | Acc↑ | Score↑ | Acc↑ | Score↑ | Acc↑ | Score↑ |
| UniVid (VAE Encoder) | 49.1 | 3.2 | 44.7 | 2.7 | 52.9 | 2.9 | 38.5 | 3.0 |
| UniVid (VAE & ViT Encoder) | 78.6 | 4.1 | 56.9 | 3.2 | 72.8 | 3.9 | 57.1 | 3.5 |
| UniVid (Ours, ViT only) | **80.1** | **4.2** | **61.4** | **3.4** | **75.0** | **4.1** | **58.8** | **3.6** |

Tab. 4 shows that removing TMA causes a noticeable drop in temporal stability, motion smoothness, and imaging quality, confirming its necessity for coherent long-horizon generation. Among different scheduling strategies, the cosine scheme consistently performs best. Its smooth transition from stronger early text guidance to later visual refinement yields better semantic fidelity and more stable dynamics than constant, step, or linear variants, highlighting the importance of a well-shaped modulation schedule.

Fig. 7 visualizes these issues: without TMA, generated players exhibit unnatural fingers, distorted poses, and implausible ball trajectories, whereas the full UniVid produces coherent jump shots with realistic ball arcs. Qualitative comparisons in Fig. 3 confirm that UniVid consistently avoids missing objects and deformations that plague prior models, achieving both semantic plausibility and temporal stability.

**Ablation on video understanding.** Tab. 7 compares four variants: a lightweight base model without our training or reasoning additions, a version w/o finetune that removes Stage-II video-QA finetuning, a version w/o Reflection that keeps finetuning but disables the Pyramid Reflection loop, and the Full UniVid. Finetuning the understanding branch on ActivityNet-QA style instruction data already yields clear gains over the base, indicating that modest, task-aligned supervision substantially improves cross-modal grounding. Adding Pyramid Reflection further boosts accuracy, with similar trends in the QA scores, confirming that query-driven keyframe selection plus the Actor–Evaluator–Reflector loop improves temporal coherence and evidence retrieval. Overall, the full system combines data-efficient tuning with iterative reasoning to deliver competitive results across all four benchmarks.

Furthermore, we investigate the impact of scaling down the Evaluator and Reflector. Specifically, we replace the originally used large-scale language model (LLM) with a more lightweight 7B LLM. As shown in Tab. 8, the results demonstrate only a marginal performance drop. This is because the primary reasoning and semantic alignment are handled by the MLLM, while the Evaluator and Reflector mainly serve to refine information selection, a process that does not heavily rely on strong reasoning capability or extensive prior knowledge. This indicates that Pyramid Reflection can be efficiently executed using smaller models, achieving a favorable trade-off between efficiency and accuracy. Notably, when we only substitute the Evaluator and Reflector with smaller LLMs while keeping the MLLM unchanged, performance degradation remains minimal, which further supports the above conclusion. Additionally, to mitigate potential understanding-evaluation(reflection) bias caused by using the same model family, we adopt different model types for Evaluator and Reflector, leading to moderate but consistent performance improvements.

Moreover, we evaluate our model on several recent benchmarks designed for unified video understanding, including MMLU (Hendrycks et al., 2021a;b), MMMU (Yue et al., 2024), MME (Fu et al., 2024a), MMBench (Fang et al., 2024), and MLVU (Zhou et al., 2024b). These datasets cover diverse multimodal reasoning tasks and reflect models' comprehensive understanding capabilities. We compare our unified model with its understanding-only models and latest Open-Source Unified Video Model to highlight our model's performance. As shown in Tab. 10, our method achieves com-

Table 10: Comparison of Und.Only and Unified Models across major video benchmarks (Hendrycks et al., 2021a;b; Fu et al., 2024a; Fang et al., 2024; Zhou et al., 2024b; Yue et al., 2024). The best results are highlighted in **bold**, and the second-best are underlined. Notably, all methods are evaluated under a unified frame-setting for fair comparison and our method can utilize **at most** unified setting frames.

| Model | MMLU↑ | MMMU↑ | MME↑ | MME(S&M)↑ | MMBench↑ | MLVU↑ |
|---|---|---|---|---|---|---|
| **Frame Num** | **32** | **32** | **64** | **64** | **64** | **64** |
| **Und.Only Models** | | | | | | |
| Qwen2-VL-7B (Wang et al., 2024a) | 21.02 | 41.26 | 59.7 | 72.1 | 1.45 | 62.34 |
| Qwen2.5-VL-7B (Bai et al., 2025) | 24.17 | 47.44 | 62.8 | 75.9 | 1.49 | 62.052 |
| Qwen3-VL-8B (Yang et al., 2025) | **71.6** | **69.9** | **71.4** | **89.7** | **2.55** | **78.1** |
| LLaVA-Video-7B (Lin et al., 2024) | 15.89 | 36.11 | 63.7 | 78.1 | 1.6 | 67.66 |
| MiniCPM-V-2.6-7B (Yu et al., 2025) | – | – | 59.7 | 74.7 | 1.7 | 52.82 |
| InternVL2_5-8B (Chen et al., 2024b) | 52.47 | 43 | 63.7 | 77 | 1.68 | 63.94 |
| InternVL3-8B (Zhu et al., 2025) | 57.71 | 47.97 | 66 | 79.5 | 1.69 | 67.964 |
| **Unified Models** | | | | | | |
| Omni-Video-7B (Tan et al., 2025) | 41.28 | 51.62 | 59.43 | 71.43 | 1.59 | 67.24 |
| Emu3-8B (Wang et al., 2024b) | 40.33 | 49.73 | 60.98 | 68.76 | 1.54 | 66.77 |
| Show-o2-7B (Xie et al., 2025b) | 45.77 | 53.99 | 66.87 | 76.62 | 1.67 | 68.92 |
| **Ours-7B** | 49.88 | 59.41 | 62.68 | 78.4 | 1.85 | 70.77 |

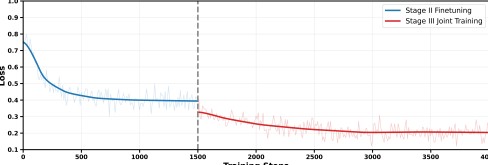

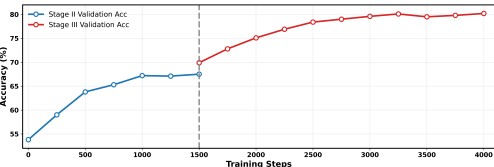

(a) Training loss curve across dual stages.    (b) Validation accuracy during training.

Figure 10: Training loss (left) and validation accuracy (right) curves for UniVid's understanding branch. Notably, red line refers to co-training period in Stage III.

petitive results on most benchmarks, particularly outperforming existing unified models. It is also worth noting that Video-MME includes longer videos (>10 min), for which we further report results under short-video (S) and mid-length (M) subsets. Our unified model shows more significant advantages on short-video scenarios, consistent with its design characteristics, while still maintaining strong overall comprehension capabilities.

**Ablation on encoding mechanism.** We study the internal encoding mechanism of UniVid. During training, we employ both a ViT and a VAE to encode visual information, where the ViT excels at capturing high-level semantics and the VAE is more effective in representing pixel-level details. We conduct ablation studies for both generation and understanding tasks to examine the role of each encoder.

For video generation, Tab. 5 shows that using only the ViT or only the VAE leads to significant degradation across almost all VBench-Long dimensions. In contrast, combining both encoders yields large improvements in overall score and boosts technical, aesthetic, and semantic fidelity metrics. This confirms that high-level semantic encoding and low-level detail encoding are complementary for long-horizon video synthesis.

For video understanding, Tab. 9 indicates that ViT alone is sufficient to achieve strong performance, while adding the VAE brings marginal or no further improvement. This aligns with the intuition that understanding tasks rely more on semantic abstraction than pixel-level reconstruction. Together, these results demonstrate that UniVid benefits from a hybrid encoding design for generation, while semantic encoders dominate in understanding.

## A.7 LIMITATION AND FUTURE WORK

While UniVid unifies an autoregressive MLLM with a DiT-based video diffusion decoder, the current interaction between the two modules remains relatively shallow. Most MLLM parameters are frozen, and the diffusion branch only receives limited semantic guidance, restricting the potential mutual benefits between understanding and generation. As a consequence, the MLLM gains little improvement in deeper reasoning, and the generation branch relies primarily on data-driven priors rather than task-aware adaptive conditioning.

These limitations manifest in characteristic failure modes during generation, as illustrated in Fig. 9 UniVid can exhibit temporal inconsistencies in long sequences (e.g., static lightning), loss of fine-grained details in distant regions, and occasional structural artifacts such as missing body parts in anthropomorphized characters. These reflect inherent challenges of long-horizon diffusion sampling and the lack of stronger semantic–structural feedback between the two branches.

In future work, we plan to develop deeper bidirectional coupling mechanisms that allow MLLM reasoning signals to shape the diffusion trajectory dynamically, while generated visual feedback reinforces semantic learning. Another promising direction is integrating native dense video encoders to support substantially longer videos with richer motion dynamics. Although these extensions require greater training resources, they offer the potential for more stable long-range generation and more emergent capabilities from cross-modal co-training.

