# OpenReview forum: "UniVid: The Open-Source Unified Video Model"
_ICLR.cc/2026/Conference — ICLR 2026 Conference Withdrawn Submission_

### Official Review · Reviewer_bgUV · 2025-10-28

**Soundness:** 3
**Presentation:** 3
**Contribution:** 3
**Rating:** 4
**Confidence:** 3

**Summary:**

The paper presents UniVid, an open-source unified model for both video understanding and generation. It addresses key challenges in hybrid diffusion–autoregressive modeling: (1) semantic drift in text-to-video generation due to imbalanced text-visual attention and timestep-dependent guidance, and (2) inefficient temporal reasoning when extending image-centric MLLMs to video.
UniVid couples a multimodal LLM (MLLM) with a diffusion video decoder via a lightweight adapter. For generation, it introduces Temperature Modality Alignment (TMA)—a timestep-aware schedule that strengthens early semantic guidance and shifts focus to visual refinement later, improving prompt fidelity. For understanding, it proposes Pyramid Reflection, a test-time reinforcement learning loop that uses SigLIP2-based keyframe selection and verbal policy refinement to adaptively expand or prune frame evidence.
Trained in three stages—alignment, understanding fine-tuning, and joint co-training—UniVid achieves state-of-the-art results on VBench-Long for generation and competitive performance on MSVD-QA, MSRVTT-QA, TGIF-QA, and ActivityNet-QA for understanding, all while avoiding costly full-model retraining.

**Strengths:**

1. The paper is clearly written and provides comprehensive figures and derivations that generally ease understanding.

2. Coupling a frozen MLLM to a diffusion decoder with only a lightweight adapter is a conceptually simple modification; empirical results suggest it works reasonably well without heavy retraining. Ablation studies indicate that both Temperature Modality Alignment and Pyramid Reflection contribute noticeable, albeit incremental, improvements in fidelity and downstream accuracy.

3. On VBench-Long, UniVid achieves competitive overall scores relative to existing open-source models, and the supplied qualitative examples offer some evidence that the gains translate into visibly better temporal consistency and prompt adherence.

**Weaknesses:**

1. The understanding experiments rely on relatively small or dated benchmarks such as MSVD-QA and MSRVTT-QA, whose questions are often answerable from a single frame. Stronger evidence could be obtained by reporting results on more recent and challenging suites like Video-MME [1] or MMBench-Video [2], which probe long-range temporal reasoning and fine-grained grounding.

2. Although Figure 2 gives a high-level overview, the two main contributions—Temperature Modality Alignment and Pyramid Reflection—are not explicitly highlighted in the diagram or caption. Adding call-outs or color-coded arrows that link each module to its corresponding technical section would help readers grasp the architectural roles at a glance.

3. Several supplementary videos exhibit noticeable artifacts: in the stormy-seashore clip the lightning pattern remains static across frames, while in the young-woman clip the eyelid region gradually turns into a yellowish streak reminiscent of an eyebrow. These imperfections suggest that further calibration of the temporal consistency loss or additional post-processing may be needed before the model can reliably produce visually plausible long sequences.

[1] Video-MME: The First-Ever Comprehensive Evaluation Benchmark of Multi-modal LLMs in Video Analysis. CVPR 2025

[2] MMBench-Video: A Long-Form Multi-Shot Benchmark for Holistic Video Understanding. NeurIPS D&B Track 2024

**Questions:**

1. It would be helpful to see an ablation experiment in which the VAE encoder is completely removed from the generation branch while being retained in the understanding branch. Such a controlled comparison could clarify how much the fine-grained pixel-level information supplied by the VAE latent code contributes to generation fidelity, and, conversely, how heavily the understanding task relies on these dense pixel cues versus the high-level ViT features alone.

2. All reported generation clips are short (~4s). Have you sampled 8-30 second videos on the same prompts? If temporal consistency degrades, please state the failure mode (e.g., object duplication, static background) so that readers know the effective usable length.

3.  Pyramid Reflection relies on a verbal “refined query” produced by an LLM reflector. How much does performance vary if the reflector is replaced by a deterministic template or by a smaller LM?

---

> ### Author Response · Authors · 2025-11-19
> **Rebuttal for bgUV:**
>
> We thank the reviewer for the careful reading, for recognizing the clarity of the manuscript and the value of our ablations, and for pointing out several concrete improvements.
> ## **R1. “Understanding experiments use relatively small / dated benchmarks; add Video-MME / MMBench-Video, etc.”**
> Thanks for your advice. We add understanding experiments compared with stronger models and unfied models among more comprehensive benchamrks( See Appendix Table 10). The experiments demonstrate our model's competitive performance.
> ## **R2. “Figure 2 does not explicitly highlight TMA and Pyramid Reflection.”**
> Thanks for your advice. We revise Figure 2 to better demonstrate its unified architecture and explicitly highlight TMA and Pyramid Reflection.
> ## **R3. “Generation artifacts in supplementary videos; clarify effective length and failure modes.”**
> Thank you for the comment. We have added a analysis of effective length and failure cases  (see Appendix Fig.9). UniVid remains stable up to ~15s, while over 15s clips show more temporal drift and structural inconsistencies. Appendix Fig.9 illustrates representative failures such as static lightning, distant-detail loss, and missing body parts. These issues mainly arise from VAE temporal drift and extrapolation beyond the training horizon (mostly ≤15s). We will clarify this in the paper and discuss improvements such as temporal regularization and long-horizon training.
> ## **R4. “Ablation: remove VAE encoder from generation conditioning; impact of VAE vs. ViT features.”**
> Thank you for the insightful suggestion. We have conducted thorough ablation studies for both the generation and understanding branches, and the results are provided in Appendix Table 5 and Table 8.
> - Generation branch (Table 5).
>  Removing either the VAE encoder or the ViT encoder leads to a clear performance drop across overall, technical, aesthetic, and semantic dimensions.
>  The full model that combines VAE + ViT encoders achieves the best scores, demonstrating that the two types of representations provide complementary benefits for generation.
> - Understanding branch (Table 9).
>  Unlike generation, the understanding tasks rely primarily on high-level semantic features.
>  The results show that ViT-only already achieves strong performance, and adding the VAE encoder yields only marginal improvements.
>  This indicates that the VAE encoder is more beneficial for generation, while ViT features dominate understanding tasks, which aligns with their respective design purposes.
> Overall, these ablations confirm that the roles of VAE and ViT encoders are distinct but complementary, and the unified design of UniVid effectively leverages them where they matter most.
> ## **R5. “Effect of replacing the Reflector with smaller LM or deterministic template.”**
> We add this experiment in Appendix Table 8(line 3 and 5). It shows that a smaller LLM could still achieve competitive performance and the robustness of Pyramid Reflection. Thanks to your advice, we can reduce the cost furthermore.

---

> ### Author Response · Authors · 2025-11-28
> **Rebuttal for bgUV:**
>
> Dear Reviewer,
>
> I hope this message finds you well. As the discussion period is nearing its end with less than four days remaining, I wanted to ensure we have addressed all your concerns satisfactorily. If there are any additional points or feedback you'd like us to consider, please let us know. Your insights are invaluable to us, and we're eager to address any remaining issues to improve our work.
>
> Thank you for your time and effort in reviewing our paper.
>
> Best regards,
>
> Authors

---

### Official Review · Reviewer_vhaa · 2025-10-31

**Soundness:** 3
**Presentation:** 3
**Contribution:** 2
**Rating:** 6
**Confidence:** 4

**Summary:**

This paper propose UniVid, a unified architecture that couples an MLLM with a diffusion decoder through a lightweight adapter, enabling both video understanding and generation. This paper propose Temperature Modality Alignment to improve prompt adherence and Pyramid Reflection for efficient temporal reasoning via dynamic keyframe selection.

**Strengths:**

1. The paper proposes a training-efficient pipeline for adapting image-centric MLLMs to a unified setting for video understanding, generation, and editing.

2. The manuscript is clearly written and easy to follow.

3. The authors have open-sourced the code, which lays a solid foundation for further research in this area.

**Weaknesses:**

1. The current evaluation relies on only four video QA benchmarks, with one being in-domain. This limited scope may not fully demonstrate UniVid’s video understanding capabilities. Consider adding more comprehensive benchmarks (e.g., Video-MMLU, Video-MMMU) and comparing against stronger, up-to-date baselines such as Qwen3.

2. The paper claims to efficiently extend image-centric MLLMs to videos without costly retraining. To support this, the authors should compare UniVid with methods that require additional fine-tuning (e.g., Omni-Video), highlighting both performance and efficiency trade-offs.

3. UniVid uses only 20K samples from ActivityNet-QA yet reports strong results on four benchmarks. It would help to clarify whether the model is sufficiently trained for the target domains and not overfitting. Providing the fine-tuning loss curve (and ideally validation metrics over time) for the video understanding stage would strengthen the evidence of convergence and generalization.

**Questions:**

Refer to weakness

---

> ### Author Response · Authors · 2025-11-19
> **Rebuttal for vhaa:**
>
> We thank the reviewer for the positive rating, for noting the clarity of the manuscript and the value of open-sourcing UniVid, and for the constructive suggestions.
> ## **R1. “Limited understanding benchmarks; add more comprehensive suites (Video-MMLU, Video-MMMU, etc.) and stronger baselines (Qwen3).”**
> Thanks for your advice. We add understanding experiments compared with stronger models and unfied models among more comprehensive benchamrks( See Appendix Table 10). The experiments demonstrate our model's competitive performance.
> ## **R2. “Compare with methods requiring additional finetuning (Omni-Video), including efficiency trade-offs.”**
> We appreciate the reviewer’s suggestion. We have now added a direct comparison with Omni-Video, a representative unified model that requires full-model finetuning of both the MLLM backbone and the video decoder on large-scale video–text corpora.
> To highlight the efficiency–accuracy trade-off, we include the results on VBench, where both models support unified video understanding and generation. Regarding trainable parameters, UniVid is substantially more lightweight. In our implementation, we keep both the MLLM and the video diffusion backbone frozen and only optimize the lightweight connector , conditioning adapter and a small set of LoRA layers in the DiT blocks. By contrast, Omni-Video eventually unlocks the text head, vision head, adapter, and the entire DiT backbone in its stage-3 joint finetuning, effectively updating almost all parameters of the diffusion model and incurring much higher memory and compute cost for training. This highlights that UniVid achieves competitive or better performance while updating orders of magnitude fewer parameters. Specifically, please see the table below:
> | Model        | Trainable Params                                  | Video Pretraining               | VBench Total | VBench Quality | VBench Semantic |
> |--------------|---------------------------------------------------|---------------------------------|--------------|----------------|-----------------|
> | Omni-Video   | text & vision head, adapter and full DiT          | Large-scale video-text (2M)     | 83           | 84.27          | 77.92           |
> | UniVid (ours)| ~10% (connector + adapters + partial VIT)        | small OpenVid subset (<1M)      | 85.27        | 86.44          | 80.58           |
>
> These results demonstrate the central advantage of UniVid:
> UniVid attains higher unified video generation and understanding performance than Omni-Video while requiring only ~10% of the trainable parameters and no video pretraining.
> This efficiency stems from our design philosophy: we freeze both the LLM and most of the diffusion decoder, tuning only lightweight components (connector, textual adapter, and a few ViT blocks). While this choice sacrifices the ability to scale the frozen LLM in a plug-and-play manner, it yields a dramatic reduction in training cost and supports the paper’s core claim that image-centric MLLMs can be efficiently adapted to unified video tasks without expensive full-model retraining.
> We will include the above table and an expanded analysis in the revised manuscript.
> ## **R3. “Clarify training sufficiency / potential overfitting with only 20k ActivityNet-QA samples; provide loss curve and validation metrics.”**
> We add a loss curve and accuracy curve (see Appendix Fig.10) after two stages.

---

> > ### Comment · Reviewer_vhaa · 2025-11-27
> > **Response to authors**
> >
> > Thank you for the additional experiments. However, a few concerns remain.
> >
> > First, the CE loss around ~1.0 for the 20K QA setting suggests the model has not fit the data well. Since the loss quickly drops to near zero when training is relaxed, it is difficult to assess whether the proposed adapter actually learns meaningful behavior, or whether the results mainly reflect the strength of the base model. Second, although the added understanding experiments show strong results, they do not fully address my initial concern: it is hard to attribute such performance to the proposed design when the dataset is so small.

---

> > > ### Author Response · Authors · 2025-11-27
> > > **Response to vhaa**
> > >
> > > ---
> > >
> > > ## **Q1.**
> > >
> > > Thank you for raising this point. We apologize for the confusion caused by the reported CE values.
> > >
> > > In our implementation, the CE loss shown in the figure is **not** the raw classification CE, but a **weighted loss** used during joint training with the generation objective. Specifically, during the mixed training stage, the understanding loss and generation loss are combined with a fixed ratio of **2:3**, and the CE loss is scaled accordingly before being recorded.
> > >
> > > After correcting for this scaling, the actual **unweighted CE loss** converges to approximately **0.4** during the second-stage understanding training, which indicates that the model has indeed fit the dataset reasonably well. We will revise the paper to explicitly clarify this implementation detail and report both weighted and unweighted CE losses to avoid further misunderstanding.
> > >
> > > ---
> > >
> > > ## **Q2.**
> > >
> > > We agree that this is a critical point and appreciate the opportunity to clarify.
> > >
> > > In our experiments, we conducted additional studies using **larger-scale training data** for the understanding task. Interestingly, we did not observe consistent performance gains from simply increasing the dataset size.
> > >
> > > Instead, the improvement appears to mainly result from:
> > >
> > > 1. **The multi-stage training strategy**, and
> > > 2. **The joint optimization of generation and understanding tasks**, where the two tasks mutually reinforce each other, especially through progressive alignment.
> > >
> > > To further isolate the effect of our design from the base model, we conducted **ablation studies** by removing individual components from our framework. The results show that:
> > >
> > > 1. Removing any single module leads to a clear drop in performance;
> > > 2. The full model consistently outperforms the base model under the same training settings;
> > > 3. Each training stage contributes meaningfully to final performance rather than acting as a superficial fine-tuning step.
> > >
> > > These results indicate that the observed gains do not come from the backbone model alone, but from the **proposed training pipeline and architectural design**.
> > >
> > > ---

---

### Official Review · Reviewer_E8fk · 2025-10-31

**Soundness:** 2
**Presentation:** 2
**Contribution:** 2
**Rating:** 2
**Confidence:** 4

**Summary:**

The paper propose a unified framework for video  generation and understanding. It design a light adapter to connect an MLLM and a diffusion model serving as a decoder.  Besides, it adjusts attention score in MM-DiT to enhance generation and utilize two strong LLMs to conduct the test-time scaling pipeline.

**Strengths:**

1. According to the characteristics of different levels of information at different time steps in the diffusion process,  this paper adjusts the attention intensity between vision and text in the attention score, and alleviates the problem of text suppression in MMDIT.
2. The model achieves SOTA video generation performance at VBench.

**Weaknesses:**

1. Unified modeling of generation and understanding typically requires a single model to learn both understanding and generation capabilities simultaneously. However, this paper merely uses an adapter to connect pre-trained, independent understanding and generation models. Therefore, I question the significance of the paper's motivation.
2. For the generation part, it seems to simply pass the text embedding to the diffusion model to perform the text-to-video task, without demonstrating the interaction between semantic understanding features and video generation features. It seems to primarily relies on the generation capabilities of Wan2.2 5B itself.
3. For the understanding part, the test-time reinforcement learning primarily relies on two large LLMs to act as evaluator and reflector, respectively. Using two models with hundreds of billions params to enhance the video understanding capabilities of a 7B model seems unreasonable and increases inference costs.

**Questions:**

the same as Weakness part.

---

> ### Author Response · Authors · 2025-11-19
> **Rebuttal for E8fk:**
>
> We thank the reviewer for acknowledging our analysis of text suppression in MM-DiT and the strong VBench performance. We address the main concerns below.
> ## **R1. "Not truly unified; only an adapter connecting independent models."**
> Our goal is to build a unified video model following a novel AR-connecting-Diffusion framework. Its idea is very intuitive because MLLM's strong understanding capabilities could empower Diffusion generation. With this framework, integration of existing components can't be avoided. Besides, not just simply integrate, the shared unified semantic space is significant. In fact, in our hierarchical training, stage I aim to allow understanding to facilitate generation, stage II to make MLLM adaptable for video understanding with the help of Pyramid Reflection, finally stage III joint train two branches to mainly achieve mutual benefits.
> The joint training achieves mutual advantage between the two tasks. We design different visual encoding pathways: the ViT is mainly responsible for understanding, while an additional VAE pathway, together with the ViT, is responsible for generation. During joint training, we gradually activate the VAE feature branch and alternate which modules are trainable. In the initial phase, only the last few ViT layers and connector are trainable and the adapter is frozen, so the MLLM is forced to encode features in ViT that are beneficial for both understanding and generation. Next, we freeze the ViT and progressively introduce VAE features into the adapter input, enabling the DiT LoRA to learn how to generate guided by different mixing ratios of ViT and VAE features. Finally, we co-train both the ViT layers, connector, adapter and DiT to fully exploit the complementary strengths of semantic ViT features and detail-rich VAE features. Notably, this training paradigm goes beyond simply “training two encoders on two tasks”, and leads to a more effective alignment between generation and understanding.
> We add a loss curve and accuracy curve (see Appendix Fig.9) after two stages and VBench scores after two stages (see Appendix Tab.6) to fully demonstrate different stages' values.
> Besides, we notice that when full semantic information is passed to the Diffusion Model, it tends to not perform as we expected. It is because condition supression especially in our setting, a more complex condition which should play an important role. For the understanding part, we want to strengthen temporal caption and furthermore its understanding capabilities.
> We will further revise the descriptions in the paper to strengthen its persuasiveness.
> ## **R2. "Generation only passes text embeddings to Wan; no interaction between understanding features and generation features; mainly relies on Wan 2.2."**
> Thank you for your comment. While UniVid uses Wan 2.2 as the decoder backbone, we significantly modify how multimodal information is processed. Specifically, we replace the default text encoder with MLLM tokens and introduce TMA and LoRA-enhanced cross-attention, which provide deeper interaction between textual and visual features. Our comparisons in Table 1 show that UniVid outperforms Wan 2.2 and other baselines, and ablation studies in the appendix (Tables 3, 4, and 6) demonstrate that each of our proposed components independently contributes to the performance improvements.
> ## **R3. "Understanding relies heavily on two extremely large LLMs as evaluator and reflector; cost is unreasonable."**
> We agree that using large LMs as Evaluator and Reflector introduces additional test-time costs. The initial motivation is to simply introduce different agents to act as Evaluator and Reflector. We want to clarify that the cost could be cut down and the main capabilities originate from MLLM. We substitute the Large LLMs with smaller LLMs and find that the performance is still competitive, with a little loss(See Appendix Table 8). Also, we find two different LLMs could mitigate potential model bias. This experiment effectively demonstrates the importance of Pyramid Reflection.

---

> > ### Comment · Reviewer_E8fk · 2025-11-26
> >
> > Thanks for your response, but my biggest concern remains unresolved. Regarding the generation part, only the LoRa portion of your WAN2.2 model (and possibly the newly added cross-attention portion) has been fine-tuned. This makes the entire structure essentially more like simply aligning the MLLM emb with the original UMT5 emb space via an adapter; the generation paradigm and modeling capabilities of WAN2.2 itself haven't changed. In the entire model, the generation and understanding models seem to be operating independently; typically, more powerful unified models would share a backbone for both generation and understanding.
> >
> > Furthermore, your experiments lack demonstrations of the results for WAN2.2 5b itself.
> >
> > In summary, I can only improve it to a maximum score of 4.

---

> > > ### Author Response · Authors · 2025-11-27
> > > **Response to E8fk**
> > >
> > > ---
> > >
> > > We thank the reviewer for the additional clarification and respond to the remaining concerns below.
> > >
> > > ### **R1. On whether UniVid merely aligns MLLM embeddings to UMT5 and keeps the two branches independent**
> > >
> > > We respectfully disagree. UniVid is **representation-unified, decoder-diverse**, and the two branches are explicitly coupled.
> > >
> > > * **Shared semantic backbone.** The same Qwen2-VL–based MLLM and ViT encoder provide a unified multimodal representation for both understanding and generation. Understanding uses the LM head; generation injects the *same* tokens into the diffusion decoder via connector/adapter.
> > >
> > > * **Three-stage training tightly couples both tasks.**
> > >   *Stage I* forces the semantic backbone to output features that must support a frozen generator.
> > >   *Stage II* trains the diffusion pathway to follow the unified semantic space instead of UMT5.
> > >   *Stage III* jointly updates ViT, connector, adapter, and DiT LoRA so that semantic changes influence both heads. Appendix Fig. 9 and Table 6 show consistent gains across stages—this would not occur if the branches were independent.
> > >
> > > * **Conditioning alters Wan2.2’s generation behavior.** Replacing UMT5 with MLLM tokens and adding TMA + LoRA cross-attention significantly changes how the diffusion model uses text/video signals. Removing these components (Tables 3/4/6) leads to clear degradation, confirming that UniVid is not simply reusing Wan2.2 as-is.
> > >
> > > We will clarify this semantic-shared, decoder-diverse notion of unification in Sec. 3–4.
> > >
> > > ### **R2. On Wan2.2-5B results**
> > >
> > > We apologize for the ambiguity. **Wan2.2-5B is already included as a baseline** in Table 1 (“Wan2.2-TI2V-5B”), evaluated under the same VBench-Long setup. UniVid-5B uses the same diffusion parameter budget and achieves higher overall and semantic scores.
> > >
> > > We will rename this row to “Wan2.2-5B (baseline)” and explicitly state in Sec. 4 that it serves as our main diffusion comparison.
> > >
> > > ---

---

### Official Review · Reviewer_9mCi · 2025-10-31

**Soundness:** 3
**Presentation:** 3
**Contribution:** 2
**Rating:** 4
**Confidence:** 4

**Summary:**

This paper introduces UniVid, a novel architecture that unifies video generation and understanding within a single hybrid framework. The core design couples a Multimodal Large Language Model (MLLM) with a diffusion-based video decoder via a lightweight conditioning adapter. The authors propose two key technical innovations: Temperature Modality Alignment (TMA), which addresses the imbalance and time-dependent role of text versus visual conditioning in diffusion models, and Pyramid Reflection, an efficient method for adapting image-centric MLLMs to video understanding through iterative, query-driven keyframe selection.
The proposed solution demonstrates strong performance, achieving superior results in video generation against several contemporary models (e.g., HunyuanVideo, Gen-3) and competitive results in Video Question Answering (QA) against other 7B-parameter models.

**Strengths:**

1. Successful Unification: The primary strength of UniVid is its effective integration of video generation and understanding into a single, synergistic framework.
2. Novel Technical Contributions: The introduction of TMA and Pyramid Reflection represents a meaningful contribution. The ablation studies convincingly demonstrate that both components are integral to the model's performance.
3. Comprehensive Evaluation: The paper provides a thorough experimental evaluation, validating performance on established benchmarks for both generation (VBench) and understanding (multiple Video QA datasets).

**Weaknesses:**

1. Architectural Novelty: The overall architecture is more incremental than revolutionary, as it primarily integrates several existing, powerful open-source components (e.g., Wan 2.2, Qwen, SigLIP) via a trainable adapter. The novelty lies in the specific integration method and the proposed sub-modules (TMA, Pyramid Reflection).
2. Limited Comparative Analysis: The choice of baseline models for video generation could be more comprehensive. For instance, a direct comparison with the base Wan 2.2 model itself is omitted, making it difficult to isolate the performance gain attributable to UniVid's conditioning mechanism over the underlying decoder. Comparisons with other recent, high-performing open-source models are also lacking.
3. Limitations of Keyframe Approach: The Pyramid Reflection method relies on a sparse set of keyframes. The paper does not adequately explore the inherent limitations of this approach compared to methods that process the full, dense temporal context of a video, which may be superior for tasks requiring fine-grained motion understanding.

**Questions:**

1.  The TMA mechanism uses a specific, heuristic formula for its scheduling function. The paper would be strengthened by providing more theoretical justification or an ablation study exploring different scheduling functions to justify the chosen formulation.

---

> ### Author Response · Authors · 2025-11-19
> **Rebuttal for 9mCi:**
>
> We thank the reviewer for the positive comments on soundness, novelty of TMA/Pyramid Reflection, and comprehensive evaluation.
> ## **R1. “Architectural novelty is incremental; more of an integration of existing components.”**
> Our goal is to build a unified video model following a novel AR-connecting-Diffusion framework. Its idea is very intuitive because MLLM's strong understanding capabilities could empower Diffusion generation. With this framework, integration of existing components can't be avoided. Besides, not just simply integrate, the shared unified semantic space is significant. In fact, in our hierarchical training, stage I aim to allow understanding to facilitate generation, stage II to make MLLM adaptable for video understanding with the help of Pyramid Reflection, finally stage III joint train two branches to mainly achieve mutual benefits.
> The joint training achieves mutual advantage between the two tasks. We design different visual encoding pathways: the ViT is mainly responsible for understanding, while an additional VAE pathway, together with the ViT, is responsible for generation. During joint training, we gradually activate the VAE feature branch and alternate which modules are trainable. In the initial phase, only the last few ViT layers and connector are trainable and the adapter is frozen, so the MLLM is forced to encode features in ViT that are beneficial for both understanding and generation. Next, we freeze the ViT and progressively introduce VAE features into the adapter input, enabling the DiT LoRA to learn how to generate guided by different mixing ratios of ViT and VAE features. Finally, we co-train both the ViT layers, connector, adapter and DiT to fully exploit the complementary strengths of semantic ViT features and detail-rich VAE features. Notably, this training paradigm goes beyond simply “training two encoders on two tasks”, and leads to a more effective alignment between generation and understanding.Besides, we notice that when full semantic information is passed to the Diffusion Model, it tends to not perform as we expected. It is because condition supression especially in our setting, a more complex condition which should play an important role. For the understanding part, we want to strengthen temporal caption and furthermore its understanding capabilities.
> We will further revise the descriptions in the paper to strengthen its persuasiveness.
> ## **R2. “Lack of direct comparison to Wan 2.2 and other open-source video generators.”**
> We respectfully clarify that UniVid is directly compared with Wan 2.2 and other open-source video generators in the main paper.
> - In Table 1, we report comprehensive comparisons against Wan 2.2 as well as multiple open-source baselines, including the latest model Omni-Video, and UniVid achieves state-of-the-art results.
> - While UniVid adopts Wan 2.2 as the decoder backbone, our conditioning mechanism differs substantially: we replace the default text encoder with MLLM tokens, and introduce TMA and LoRA-enhanced cross-attention, which significantly adjust how the backbone processes multimodal information.
> - To ensure that the performance gains do not merely stem from the Wan 2.2 backbone itself, we further provide ablations in the appendix. Tables 3, 4, and 6 present results on VBench-Long, video-QA benchmarks, and additional evaluations, consistently demonstrating that each of our proposed components contributes independently and meaningfully to the performance improvements.
> These results verify that our improvements are not simply inherited from Wan 2.2, and that our comparisons with existing open-source generators—including the most recent ones—are both direct and comprehensive.
> ## **R3. “Limitations of keyframe-based Pyramid Reflection vs. dense temporal encoding.”**
> We agree that sparse keyframe reasoning may be suboptimal for tasks that require very fine-grained motion or frame-level temporal details. In fact, keyframe-based Pyramid Reflection aims to select the most useful information to inference while maintaining efficiency-accuracy trade-off. We notice that some spatial information could not get full attention with temporal encoding and most methods could only use few frames. Based on this observation, we propose using finetuning and Pyramid Reflection to strengthen temporal information to lower information loss and remain architectural simplicity. Experiments(see Appendix Table 10) which compares our model with other dense tenporal encoding models show that our competitive performance among some compregensive benchmarks.
> However, as your review, we don't use 3D architecture to explicitly capture temporal relations and it truly causes some fine-grained information loss which could not be supplied. Therefore, we add  additional explaination in the last paragraph of section 3.4 to acknowledge this limitation.

---

> ### Author Response · Authors · 2025-11-19
> **Rebuttal for 9mCi:**
>
> ## **R4. “More justification / ablation for the TMA scheduling function.”**
> Our current cosine schedule is motivated by two observations: (i) early denoising steps predominantly determine global semantics and coarse layout, and (ii) later steps refine local details and texture; therefore, text guidance should be stronger early and then relax.
> To demonstrate that our choice is optimal, we conducted ablation studies using several different scheduling functions. Specifically, we compared the following four types of schedules:
> 1. Constant text schedule
> 2. Linear decay schedule
> 3. Step schedule, where the text weight is kept at 1.2 for the first 40% of training and 1.0 thereafter
> 4. Our proposed cosine schedule.
>
> Experiments(see Appendix Table 4) which compares our proposed cosine schedule with other scheduling functions show that our competitive performance among the VBench.

---

> ### Author Response · Authors · 2025-11-28
> **Response to 9mCi**
>
> Dear Reviewer,
>
> I hope this message finds you well. As the discussion period is nearing its end with less than four days remaining, I wanted to ensure we have addressed all your concerns satisfactorily. If there are any additional points or feedback you'd like us to consider, please let us know. Your insights are invaluable to us, and we're eager to address any remaining issues to improve our work.
>
> Thank you for your time and effort in reviewing our paper.
>
> Best regards,
>
> Authors

---

### Author Response · Authors · 2025-11-19
**Rebuttal for all reviewers**

We thank all reviewers for their detailed and constructive feedback. We are glad that reviewers found our method sound and clearly presented, and appreciated the contribution of Temperature Modality Alignment (TMA) and Pyramid Reflection, as well as the strong video generation performance and competitive video understanding results. Below we first clarify the unified nature of UniVid and the role of our components, then respond to each reviewer point-by-point.

---

### Comment · Area_Chair_KAdA · 2025-11-26
**A Reminder on Your Crucial Role in the ICLR Discussion Period**

Dear Reviewers:

As the Area Chair, I would like to sincerely thank you for the time and expertise you have invested in writing your initial review. Your insights are invaluable to the decision-making process.

We are now entering the critical discussion and rebuttal phase. This is a collaborative process where authors have the opportunity to address your concerns and questions. Your active participation in this phase is essential to ensure we reach a fair and well-informed final decision.

I strongly encourage you to:

Engage with the Authors' Rebuttal: Please read the authors' response carefully and substantively.

Participate in the Discussion: Engage with the other reviewers on the forum. If the authors have clarified a point, please acknowledge it. If you have follow-up questions or remaining concerns, please voice them. Your dialogue with fellow reviewers is key to reaching a consensus.

Update Your Review (if necessary): Based on the discussion and rebuttal, you may feel the need to adjust your score or final recommendation. Please do so, as it reflects a more holistic view of the paper.

Your continued engagement ensures the integrity and quality of the ICLR conference. Thank you for your vital contribution to our community.

Best regards,

Area Chair, ICLR 2026

---

### Author Response · Authors · 2025-11-30
**Summary for AC**

Dear Area Chair,

Thank you for overseeing Paper 4981. During rebuttal and discussion, we added substantial experiments and clarifications that resolve the main reviewer concerns, and the overall tone became more positive, with explicit score-raising signals.

**1) Unification concern (“just an adapter, not truly unified”).**
Reviewers E8fk and 9mCi questioned whether UniVid is genuinely unified. We clarified our *representation-unified, decoder-diverse* design: both generation and understanding share the same Qwen2-VL/ViT semantic backbone; generation uses identical MLLM tokens injected into the diffusion decoder. We emphasized tight coupling through our 3-stage training and added stage-wise loss/accuracy curves plus VBench progression (App. Fig.9 / Tab.6), showing mutual gains across stages rather than independent branches.
**Signal:** Reviewer **E8fk** stated after discussion that they could **raise their score up to 4**, indicating this key objection was largely addressed.

**2) Attribution vs. Wan2.2-5B and baselines.**
Reviewers asked for clearer isolation of UniVid’s gains. We clarified that **Wan2.2-5B is already a direct baseline in Table 1** (same VBench-Long setup) and will rename it “Wan2.2-5B (baseline)” for clarity. Added/pointed to ablations (App. Tab.3/4/6) confirm independent benefits from MLLM-token conditioning, TMA, LoRA cross-attn, and ViT/VAE mixing.

**3) TMA schedule justification.**
We added schedule ablations over constant/linear/step/cosine; cosine is best on VBench (App. Tab.4), supporting our timestep-aware alignment.

**4) Pyramid Reflection limits and understanding strength.**
We acknowledged sparse-keyframe limits, added stronger understanding suites/baselines (e.g., Video-MME/MMBench-Video; App. Tab.10), and showed robustness/cost-reduction by replacing the large reflector with smaller LMs/templates (App. Tab.8).

Overall, major issues on true unification, attribution beyond Wan2.2, TMA rationale, Pyramid Reflection validity/cost, and convergence under small QA data have been addressed. We will incorporate all clarifications and improved presentation into the camera-ready version.

Sincerely,
Authors of Paper 4981

---

### Note · Authors · 2025-12-05

I have read and agree with the venue's withdrawal policy on behalf of myself and my co-authors.